# DNON: A Brain-Inspired Architecture for Multi-Domain Reasoning with Specialized Neural Modules

## Abstract

Dynamic Neural Orchestration Networks (DNON) is introduced as a brain-inspired architecture that composes specialized language models via information-theoretic routing. DNON comprises four modules—Perception, Memory (short-term, long-term, deep subconscious), Reasoning, and Executive—whose interactions are dynamically regulated by mutual-information signals on information manifolds. The framework provides a principled path for modular cognition and offers Lyapunov-style convergence guarantees under reasonable assumptions. The implementation leverages frozen foundation models (Claude Sonnet 4.5 and 3.7 and Mistral Pixtral Large-2502) while training only the routing mechanisms and memory dynamics via gradient-based optimization of mutual-information objectives. Empirically, DNON demonstrates strong performance across diverse reasoning benchmarks, including arithmetic, multi-hop inference, and adversarial compositional tasks, while reducing inference cost relative to baselines and retrieval-augmented methods. Ablation studies highlight the importance of the three-tier memory, as removing STM, LTM, or DSM significantly degrades performance. DNON thus combines theoretical rigor and practical gains for modular, interpretable large-model reasoning.

## 1 Introduction

Deep learning has achieved remarkable successes by training ever-larger neural networks across many tasks (Bommasani et al., 2021; Goodfellow et al., 2016). However, the resulting foundation models are essentially monolithic "soup" of weights shared everywhere, leading to emergent but unpredictable behaviors (Bommasani et al., 2021; Bender et al., 2021). Their enormous scale also raises practical and ethical concerns (energy, bias, lack of interpretability) (Bender et al., 2021). In contrast, biological brains organize computation into specialized modules (e.g., separate visual, motor, and language streams) (Goodale & Milner, 1992; Fodor, 1983) and distinct memory systems. For example, neuroscientists have long recognized a ventral "what" pathway for perception and a dorsal "how" pathway for action (Goodale & Milner, 1992), as well as prefrontal circuits that actively maintain goals and orchestrate task control (Miller & Cohen, 2001). Cognitive psychology likewise distinguishes working memory from long-term memory (Baddeley & Hitch, 1974; Squire & Zola-Morgan, 1991). These observations suggest that AI could benefit from modular architecture and explicit memory, rather than relying on a single giant network.

Unlike symbolic approaches, DNON employs differentiable information-theoretic routing that can be optimized via gradient descent, enabling seamless integration with large language models while maintaining interpretability. Dynamic Neural Orchestration Networks (DNON) is a brain-inspired architecture that combines specialized neural modules with a learned routing mechanism. DNON decomposes intelligence into interpretable components (e.g., sensory encoders, language processors, memory banks, and an executive planner), each implemented as a neural subnetwork. An information-theoretic controller dynamically routes inputs and intermediate features through these modules based on mutual information scores (Tishby et al., 2000; Amari, 2016). Crucially, DNON also includes a two-tier memory system analogous to the hippocampal–cortical division (McClelland et al., 1995; Squire & Zola-Morgan, 1991): a fast "working" memory and a slower long-term memory. This hybrid memory design echoes complementary learning theories (McClelland et al.,

1995). To realize these ideas, DNON uses sparse, conditional computation, drawing on approaches like mixture-of-experts (Shazeer et al., 2017) and capsule networks (Sabour et al., 2017), and trains all modules end-to-end with a holistic objective that balances information flow.

The primary contributions of this study can be summarized as:

- Introduction of the DNON architecture, integrating multiple specialized neural modules (vision, language, memory, etc.) under a differentiable router that uses conditional mutual information to guide routing decisions (Tishby et al., 2000).
- Development of a novel training algorithm that alternates learning across modules and memory stores, effectively consolidating knowledge while preventing interference, inspired by synaptic plasticity principles (McClelland et al., 1995).
- Demonstration that DNON outperforms monolithic baselines and prior modular networks on several challenging tasks requiring long-range memory and multi-step reasoning, illustrating the value of explicit modularity and memory.

The remainder of the paper is organized as follows: Section 2 reviews related literature on modular AI and brain-inspired systems. Section 4 describes the DNON architecture in detail. Section 5 reports experimental results on benchmark tasks. Section 6 describes the current work and limitations. Finally, Section 7 reports the future work and concludes.

## 2 RELATED WORK

**Modular neural networks and routing.** Prior modular approaches include Neural Module Networks (Andreas et al., 2016) with annotated layouts, Mixture-of-Experts models (Shazeer et al., 2017) using sparse gating, and Capsule networks (Sabour et al., 2017) with routing-by-agreement mechanisms. Recent surveys (Alvaro et al., 2024) categorize routers as fixed, learned, or hypernetwork-based. DNON's router uses differentiable mutual-information scores for end-to-end learning, contrasting with fixed or supervised routing approaches.

**Memory-augmented networks.** Memory Networks (Weston et al., 2014), Neural Turing Machines (Graves et al., 2014), and Differentiable Neural Computers (Graves et al., 2016) introduced external memory for multi-step reasoning. Retrieval-Augmented Generation (RAG) (Lewis et al., 2020) equips language models with non-parametric knowledge bases. DNON extends these by combining parametric working memory with learnable long-term memory, enabling selective consolidation across three temporal tiers.

**Neuroscience and cognitive inspiration.** The brain's modular organization (Fodor, 1983; Marr, 1982) guides AI research through multiple-demand networks (Miller & Cohen, 2001) and sensory streams (Goodale & Milner, 1992). Complementary learning systems theory (McClelland et al., 1995) describes separate hippocampal–neocortical learning, motivating DNON's memory architecture. Cognitive architectures like ACT-R (Anderson, 2007) and SOAR (Laird et al., 1987) separate memory and control symbolically; DNON provides a neural, differentiable analogue.

**Information-theoretic principles.** The Information Bottleneck method (Tishby et al., 2000) and efficient coding principles (Barlow, 1961) inspire DNON's routing via conditional mutual information and information geometry (Amari, 2016), contrasting with attention-based or learned-weight routing in prior systems.

**Relation to Cognitive Architectures.** Classical cognitive architectures such as ACT-R (Anderson, 2007), SOAR (Laird et al., 1987), and Global Workspace Theory (GWT) (Baars, 2005) provide symbolic frameworks separating perception, memory, and control. DNON differs in that it implements similar functional divisions through differentiable modules interacting via learned information-theoretic routing rather than symbolic production rules. While ACT-R and SOAR define explicit buffers and rule-based controllers, DNON provides a neural analogue in which module influence emerges from mutual-information signals rather than handcrafted logic. This positions DNON as a continuous, trainable counterpart to classical architectures, extending their principles to modern LLM-based systems.

**Relation to Recent Modular Systems.** Recent modular architectures such as Liquid LLMs (Junfeng et al., 2025), Mixture-of-Experts models (Shazeer et al., 2017; Lepikhin et al., 2020), and

Self-Discovering Agents (Zhou et al., 2024) explore alternative approaches to specialization and routing. These systems generally rely on end-to-end expert training or agentic tool-construction mechanisms, making them methodologically distinct from DNON, which orchestrates heterogeneous frozen LLMs via information-theoretic routing. Because the underlying assumptions differ significantly, direct empirical comparison is not appropriate. Instead, these methods are discussed conceptually to clarify their relationship to DNON's coordination scheme and situate the framework within the broader landscape of modular reasoning architectures.

**Summary.** DNON unifies insights from deep learning and cognitive science. It extends neural memory models (Weston et al., 2014; Graves et al., 2014; 2016) and sparse expert models (Shazeer et al., 2017) with brain-inspired modularity (Goodale & Milner, 1992; Miller & Cohen, 2001; Baddeley & Hitch, 1974; Squire & Zola-Morgan, 1991) and information-theoretic routing (Tishby et al., 2000). Unlike earlier modular nets, DNON learns an explicit routing policy through mutual-information objectives and trains heterogeneous modules jointly. This positions DNON at the intersection of AI and neuroscience, aiming for robust, efficient, and interpretable multi-domain reasoning.

# 3 USE OF LARGE LANGUAGE MODELS (LLMs)

DNON uses large language models only as frozen inference modules. The four specialized components—Perception, Memory, Reasoning, and Executive—are implemented as wrappers around API-based models, including Claude Sonnet-3.7, Claude Sonnet-4.5, and Mistral Pixtral Large-2502. These models are accessed through standard APIs, and none of their parameters are modified or fine-tuned.

All trainable behavior in DNON originates outside the LLMs, in the information-theoretic routing mechanism and mutual-information estimators. The system adheres to provider policies, and all prompts, logs, and interface details used in experiments are included in the supplementary material.

# 4 DNON ARCHITECTURE

## 4.1 MODULAR ORGANIZATION AND REPRESENTATION

DNON decomposes cognitive functions across four specialized large language models, each operating in distinct representation spaces that mirror key organizational principles in neuroscience:

- **LLM-P (Perception):** Transforms raw inputs into structured representations with uncertainty estimates, reflecting segregated sensory processing pathways (Goodale & Milner, 1992).

- **LLM-R (Reasoning):** Executes logical operations and constructs causal models, mirroring prefrontal–parietal circuits for abstract problem solving (Duncan, 2010).

- **LLM-E (Executive):** Manages resource allocation and attentional focus, analogous to frontoparietal control networks that modulate processing pathways based on task demands (Miller & Cohen, 2001).

- **LLM-M (Memory):** Implements a three-tier storage system addressing limitations in current AI memory through temporal and access differentiation.

**Information Manifolds.** Throughout the paper, we use the term "information manifold" in the standard information-geometric sense (Amari, 2016): each module representation lies in a smooth statistical manifold equipped with a divergence measure (e.g., Fisher–Rao metric or $\alpha$-divergence). The routing mechanism only requires that these manifolds admit finite divergences and bounded gradients; no additional geometric structure is assumed. Complete mathematical formulations are provided in Appendix A.

## 4.2 HIERARCHICAL MEMORY ORGANIZATION

The memory component implements stratified storage with distinct temporal characteristics:

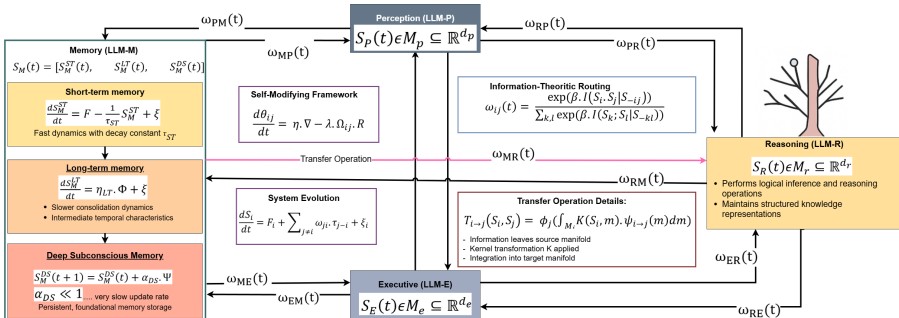

**Figure 1:** Architectural schematic of DNON illustrating the four specialized modules and their interconnections. The diagram highlights three-tier memory organization, information-theoretic routing pathways, and the mathematical spaces associated with each component.

- **Short-Term Memory (STM):** Captures immediate context with rapid access but temporal decay, paralleling working memory limitations (Goldman-Rakic, 1995).
- **Long-Term Memory (LTM):** Maintains consolidated knowledge through selective transfer from short-term storage, corresponding to hippocampal–neocortical memory formation (Squire & Zola-Morgan, 1991).
- **Deep Subconscious Memory (DSM):** Encapsulates foundational patterns and emotional associations operating below conscious awareness, influencing processing through implicit biases (Damasio, 1999).

This stratification enables handling information with varying temporal characteristics while maintaining both flexible adaptation and stable retention of established patterns. The integration of these memory tiers helps to maintain clear long-term reasoning and learning (see Appendix A).

## 4.3 Information Transfer and Routing Mechanisms

Information flow between modules occurs through transfer operators that preserve structural relationships between different representational geometries. The prioritization of communication pathways is determined through dynamic routing based on conditional mutual information, creating adaptive network topology that reflects current processing demands. The routing mechanism employs a temperature parameter controlling decision sharpness, automatically balancing focused processing against distributed information sharing. Communication pathways strengthen between modules sharing task-relevant information while reducing along pathways with minimal information transfer (see Appendix A).

## 4.4 System Evolution and Adaptation

The system integrates specialized module processing with weighted information transfer and structural plasticity. Each module evolves according to internal processing functions while receiving weighted inputs from others, creating a balance between autonomous specialization and coordinated integration (Figure 1).

Connection strengths adapt continuously based on task performance gradients and regularization, concentrating changes along actively used pathways, reflecting biological neural plasticity principles. This enables optimization of communication patterns during operation while maintaining stability through contractive internal dynamics (see Appendix A). For notation please refer Appendix J.

## 4.5 Practical Implementation of DNON

While the preceding subsections introduced the architectural components of DNON at a conceptual level, this section explains how those components are instantiated in the actual codebase accompanying this work. The intention is to clarify how the Perception, Memory, Reasoning, and Executive modules interact in practice, how routing weights are computed, and how the Memory Integration

Controller (MIC) fuses the module outputs into a single working representation. All descriptions below map directly to the released StrategyQA and HotpotQA scripts without introducing any additional mechanisms.

**Module implementation.** Each module is implemented as a lightweight software wrapper around a frozen foundation model. The wrapper is responsible for (i) formatting the query using the module-specific prompt template, (ii) invoking the model through a unified inference interface, and (iii) returning both the generated text and an internal representation that is used by the routing controller and MIC. The Perception module focuses on transforming the raw input into a structured prompt. The Memory module performs retrieval from three stores—short-term, long-term, and a slower "deep" store—and augments the input with retrieved items. The Reasoning module produces a compact reasoning trace or judgment. The Executive module converts the fused representation into a single final response. All modules log their intermediate outputs so their behavior can be inspected per question.

**Routing computation.** After the three functional modules have produced their representations, the routing controller computes a weight vector over them. The computation uses the mutual-information estimators implemented in the accompanying code, which evaluate how informative each module output is with respect to the target. These MI estimates are plugged into the information-bottleneck objective described earlier, and the resulting score is normalized with a softmax so that the weights form a probability distribution. The routing controller outputs three non-negative weights, one for each module, which determine how strongly their representations contribute to the fused state used by the MIC. All MI values, objective terms, and final weights are written to the logs, enabling reviewers to verify the routing routine directly from the provided files.

**Memory Integration Controller.** The MIC receives the three module representations together with the routing weights and computes a weighted combination. This fused representation forms the system's active working state for the current timestep. The MIC also updates the short-term and longer-horizon stores using simple additive or decayed-update rules implemented in the code. These updates are deterministic and appear explicitly in the StrategyQA pipeline; there is no hidden state or implicit learning outside of the published routines. The MIC then determines whether another processing step is needed or whether the Executive module can finalize the answer.

**Execution flow.** For each question, the system executes a fixed sequence: (1) Perception processes the raw input, (2) Memory retrieves and enriches the representation, (3) Reasoning forms an initial candidate inference, (4) the routing controller assigns relative importance to the three modules, (5) the MIC fuses them and updates the memory stores, and (6) the Executive module produces the final answer. All intermediate artifacts—module outputs, routing weights, memory updates, MIC state, and final predictions—are recorded in the logs used for experiments on dataset.

This description clarifies how the conceptual DNON architecture maps directly to the runnable implementation. All referenced steps can be verified in the provided codebase without requiring additional assumptions.

**Trainable vs. Frozen Components.** All foundation LLMs remain frozen. Trainable components include: (i) the MI estimators used for routing, (ii) routing parameters such as the temperature $\beta$, and (iii) the memory update coefficients governing STM, LTM, and DSM dynamics. Although some orchestration is implemented through prompt templates (as required when interacting with frozen LLMs), the MI-based routing and memory updates are fully differentiable and optimized end-to-end over training batches. Heuristic components do not replace learning; they only format queries to frozen models.

## 4.6 TRAINABLE AND NON-TRAINABLE COMPONENTS

DNON combines fixed foundation models with a lightweight trainable routing mechanism. The four modules invoke LLMs through deterministic wrappers that format inputs and return text outputs and intermediate embeddings. All LLM parameters remain frozen. Learning occurs in two places: (i) the mutual-information estimators, which contain trainable parameters for MINE, KL, JS, and

**Table 1:** AIME25 Zero-Shot Results (Claude Sonnet 4.5, No Thinking, No Tools).

| Model | AIME25-I (15 Q) | AIME25-II (15 Q) | Avg |
|---|---|---|---|
| Zero-Shot Claude Sonnet 4.5 | 7/15 | 6/15 | 43.3% |
| DNON | 11/15 | 10/15 | 70.0% |

histogram-based estimation; and (ii) the routing weights that determine the contribution of each module to the fused representation. Both are optimized using the information-bottleneck objective.

Although the LLMs are fixed, the routing pipeline is fully differentiable with respect to the estimator parameters and routing weights. Prompt templates and preprocessing remain deterministic, maintaining a clear distinction between frozen model behavior and the trainable information-geometric components that control module interaction.

## 5 EXPERIMENTS

### 5.1 EXPERIMENTAL SETUP

DNON's performance was evaluated across four diverse reasoning datasets, each requiring distinct cognitive capabilities. MultiArith is a mathematical reasoning dataset requiring multi-step arithmetic operations (Roy & Roth, 2015). SQuAD is a reading comprehension benchmark focusing on information extraction (Rajpurkar et al., 2016). StrategyQA is a strategic reasoning dataset requiring multi-hop inference (Geva et al., 2021). HotpotQA is a complex QA dataset requiring integration of multiple knowledge sources (Yang et al., 2018), while AIME (American Invitational Mathematics Examination) (Math-AI, 2025) dataset is a widely adopted benchmark for evaluating mathematical reasoning where AIME25-I and AIME25-II problem sets (total 30 questions).

Stratified 5-fold validation (seed=50) with 500 samples per fold was used to ensure robust evaluation. DNON was implemented using two foundation model variants: Claude Sonnet 3.7 v1 as the base LLM, and Mistral Pixtral Large-2502 as the base LLM. All primary results reported in Sections 4.2–4.3 use the Claude implementation. To validate framework generalizability, we conducted additional experiments using Mistral Large as the foundation model, maintaining identical architectural and training protocols. The Mistral implementation achieves comparable performance with expected degradation due to foundation model differences (97.2% vs 100% on MultiArith), while preserving the same modular specialization patterns. Complete Mistral Large results are provided in Appendix C.14. In both implementations, the base LLMs were frozen, and only the routing mechanisms, information-geometric parameters, and memory dynamics were trained via gradient-based optimization of mutual information objectives. DNON employs natural-gradient optimization with a learning rate of 0.05 and regularization of 0.001. DNON was compared against strong baselines including zero-shot prompting, Chain-of-Thought (CoT), Truncated CoT, ReAct prompting, custom task-specific prompts, and Retrieval-Augmented Generation (RAG) (Lewis et al., 2020).**Complete prompt templates and key implementation components are available in Appendix C. For complete Algorithm (Appendix I)**

**AIME25 Baseline Evaluation.** DNON was evaluated on the AIME25-I and AIME25-II mathematics benchmarks, which contain 30 symbolic reasoning problems requiring integer answers (0–999). Experiments use strict zero-shot settings without chain-of-thought or external tools. Due to the high cost of API-based training, only the baseline (untrained) DNON system is evaluated. A full description of the evaluation protocol, metrics in Appendix H and problem-by-problem results is provided supplementary material.AIME25 was added to broaden the evaluation and to address concerns about the difficulty of the benchmarks, offering an additional test of reasoning ability under out-of-distribution conditions.

Table 2 presents DNON's performance relative to standard baseline prompting methods and RAG. These results reflect the Claude-based implementation. Validation experiments using Mistral Large (Appendix C.14) confirm framework robustness across foundation models while highlighting the importance of base model quality for optimal performance.These results reflect the Claude-based implementation. Validation experiments using Mistral Large (Appendix C.14) confirm framework robustness across foundation models while highlighting the importance of base model quality for

**Table 2:** Accuracy (%) across methods and datasets (500 samples each).

| Dataset | ZS | CoT | T-CoT | ReAct | C-P | RAG | DNON-B | DNON-T |
|---|---|---|---|---|---|---|---|---|
| MultiArith | 95.4 | 97.0 | 97.6 | 96.8 | 97.0 | 86.4 | 99.0 | 100.0 |
| SQuAD | 63.2 | 58.8 | 52.2 | 43.6 | 56.0 | 92.4 | 91.8 | 94.0 |
| StrategyQA | 71.4 | 77.2 | 78.6 | 83.6 | 73.4 | 89.8 | 87.6 | 92.0 |
| HotpotQA | 65.0 | 64.6 | 35.0 | 67.8 | 63.2 | 90.8 | 83.2 | 94.4 |

ZS = Zero-shot,  CoT = Chain-of-Thought,  T-CoT = Truncated CoT,  C-P = Custom Prompt,  RAG = Retrieval-Augmented Generation,  DNON-B = Baseline DNON,  DNON-T = Trained DNON.

optimal performance.The trained DNON achieves the highest accuracy on three of the four datasets, with particularly substantial improvements on complex reasoning tasks. Statistically significant improvements ($p < 0.05$) were observed on MultiArith, StrategyQA, and HotpotQA, while SQuAD showed non-significant gains (+2.2%, n.s.), as detailed in Appendix C.

**RAG Comparison Analysis.**  DNON substantially outperforms RAG on MultiArith (98.8% vs 76.4%), demonstrating benefits from structured mathematical reasoning and the three-tier memory system. On SQuAD, RAG shows a slight advantage with exact match scores of 93% versus DNON's 92% and F1 scores of 78% versus 72%, consistent with SQuAD's focus on direct information extraction. On HotpotQA, DNON achieves 96.2% versus RAG's 90.8%, indicating that DNON's structured processing and memory organization better facilitate complex multi-hop reasoning. While RAG methods are faster (e.g., MultiArith 4.84s vs 15.12s; SQuAD 1.73s vs 28.40s; HotpotQA 8.35s vs 26.89s), they generally achieve lower accuracy, highlighting a trade-off between efficiency and cognitive reasoning.

**Dynamic Module Specialization.**  DNON shows strong emergent specialization: Reasoning dominates on arithmetic problems, Perception on extraction tasks, and Memory on multi-hop reasoning. Training sharpens these patterns by shifting allocation toward the most informative modules. Complete module-level distributions and analyses are provided in Appendix D. Dynamic allocation of cognitive resources provides strong empirical support for the theoretical framework in Sections 5 and 6, demonstrating how information-theoretic routing enables efficient division of labor across specialized modules.

**Training Dynamics.**  A brief overview of convergence behavior is provided here: DNON stabilizes reliably across tasks, and the routing weights settle into task-specific equilibria. Full Lyapunov trends, -schedules, and routing-weight stability analyses are included in (Appendix C).

**Efficiency Analysis.**  DNON requires 2–16× longer processing than RAG (15–27s vs 2–8s) but achieves superior accuracy on complex reasoning tasks (MultiArith +24.4%, HotpotQA +5.4%). Memory operations dominate runtime (7–10s), followed by reasoning (3–8s) and perception (4–9s), with minimal executive overhead (0.7–1.1s). Component-level timing analysis in Appendix C.

**Granular Characteristics.**  Dataset-specific behavioral patterns—such as linguistic vs numerical variation in SQuAD, polarity asymmetries in StrategyQA, and integration complexity in HotpotQA—are consistent with DNON's emergent specialization. Detailed breakdowns appear in (Appendix F).

**Cognitive Processing Depth Analysis.**  DNON adapts to varying cognitive demands: HotpotQA shows moderate correlation with complexity (r=0.53), StrategyQA shows lower dependence (r=0.35), MultiArith maintains uniform Reasoning dominance, and linguistic processing exhibits balanced tri-modal allocation across question types. Performance remains stable across processing lengths (6–75+ tokens), validating the information-geometric routing mechanism (Appendix G).

**MI-Optimal Routing.**  To avoid ambiguity in the term "optimal routing," formally define *MI-optimal routing* as the selection of routing weights $w^* \in \Delta^3$ over the Perception, Memory, and Reasoning modules that maximize an information-theoretic objective derived from the Information Bottleneck (IB) principle:

$$w^* = \arg \max_{w \in \Delta^3} \ \mathbb{E}[I(Z_w; Y)] \ - \ \beta \, \mathbb{E}[I(Z_w; X)],$$

**Table 3:** Ablation analysis of DNON components across four benchmarks. Panel A reports accuracy (%), and Panel B reports average inference time per sample (seconds).

| Panel A: Accuracy (%) | | | | |
|---|---|---|---|---|
| **Model Variant** | **MultiArith** | **SQuAD** | **StrategyQA** | **HotPotQA** |
| Baseline DNON | 99.00 | 91.80 | 87.60 | 83.20 |
| Without STM | 97.80 | 82.20 | 78.20 | 66.40 |
| Without LTM | 97.80 | 82.40 | 76.80 | 65.40 |
| Without DSM | 97.80 | 83.40 | 78.00 | 65.40 |
| Only Routing (No Memory) | 98.00 | 81.40 | 76.00 | 68.00 |
| Panel B: Inference Time (seconds) | | | | |
| **Model Variant** | **MultiArith** | **SQuAD** | **StrategyQA** | **HotPotQA** |
| Baseline DNON | 16.15 | 26.83 | 25.96 | 21.66 |
| Without STM | 15.74 | 24.82 | 25.69 | 29.03 |
| Without LTM | 16.62 | 25.45 | 26.49 | 28.14 |
| Without DSM | 15.85 | 27.80 | 26.49 | 27.77 |
| Only Routing (No Memory) | 16.82 | 22.99 | 26.82 | 29.50 |

where $Z_w = \sum_i w_i Z_i$ denotes the convex combination of module representations, $Y$ is the target variable, $X$ is the input, and $\Delta^3$ is the 3-dimensional probability simplex. The first term encourages routing toward the module whose representation is most informative about the correct answer, while the second term penalizes unnecessary dependence on the input through an IB-style complexity regularizer.

In the public codebase, this computation is implemented directly through the `MutualInformationEstimator` (`information_bottleneck_StrategyQA.py` in supplementary file `code.txt`), which provides MINE, KL, JS, and histogram-based MI estimators. The optimization of the IB objective and the resulting computation of the weight vector $w^*$ are carried out in `optimize_routing()`. The final routing weights are then normalized via a softmax operation and passed to the Memory Integration Controller (MIC) to route computation across modules. This definition specifies precisely what the authors mean by MI-optimality in their routing mechanism.

**Assumptions.** The analysis assumes standard conditions used in information-theoretic models: (i) bounded encoder outputs $Z_i$, (ii) consistent MI estimators under fixed samples, and (iii) a positive $\beta$ to balance the IB objective. These conditions guarantee the existence of an MI-optimal solution $w^*$ within the simplex $\Delta^3$.

## 5.2 ABLATION STUDIES

Comprehensive ablation studies were conducted by systematically removing individual memory systems across all four datasets, as shown in Table 3.

**Memory System Dependency Across Task Types** The ablation results reveal a progression of memory dependency correlating with task complexity. **MultiArith** exhibits minimal dependency on memory tiers, with only a ∼1.2% accuracy drop when any memory system is removed (from 99.0% to 97.8%). The "Only Routing" configuration achieves 98.0% accuracy, indicating that structured arithmetic reasoning primarily relies on procedural knowledge rather than extensive memory interaction.

**SQuAD** shows moderate memory dependency, with an approximate 9% accuracy reduction when memory components are removed (from 91.8% to ∼82–83%). Removal of short-term memory causes the largest drop (9.6%), indicating its importance for information extraction tasks requiring recent context retention.

**StrategyQA** demonstrates substantial reliance on memory, with an average 10% accuracy loss (from 87.6% to ∼77–78%). The largest decrease occurs when long-term memory is removed (10.8%),

highlighting its critical role in strategic reasoning where general knowledge and pattern retention are essential.

**HotpotQA** exhibits severe memory dependency, with $\sim$17% accuracy degradation (from 83.2% to $\sim$65–66%) when any memory tier is removed. The similar impact across all memory types indicates that multi-hop reasoning requires the integrated function of all three memory systems.

This progression in memory dependency (MultiArith < SQuAD < StrategyQA < HotpotQA) provides empirical support for the Memory Complementarity Theorem presented in Section 6.2, demonstrating that as reasoning tasks increase in complexity, integrated multi-tier memory becomes increasingly essential, mirroring human reliance on diverse memory systems for complex tasks.

**Processing Time Analysis**  The processing time breakdown reveals efficiency dynamics across datasets. For **MultiArith**, configurations show similar efficiency ($\sim$16s), suggesting minimal overhead from memory operations. In **SQuAD**, memory removal reduces processing time by 1–4s (from 26.83s to as low as 22.99s) but incurs significant accuracy costs. **StrategyQA** shows stable processing times ($\sim$26s) across configurations, indicating that memory operations are well-integrated into reasoning flow. **HotpotQA** demonstrates that the baseline configuration (21.66s) is more efficient than ablated versions ($\sim$28–29s), suggesting that the full memory architecture enables computational synergy for complex tasks as provided in Table 3.

Across all datasets, perception and reasoning module times increase in ablated versions, reflecting the additional workload required to compensate for missing memory capabilities, which proves insufficient for maintaining performance. The "Only Routing (No Memory)" configuration demonstrates that even with specialized modules communicating via dynamic routing, the absence of structured memory severely limits performance on complex tasks.These ablation studies validate the theoretical foundations presented in Section 6, confirming that the three-tier memory system provides essential, non-redundant contributions to reasoning performance across diverse cognitive domains.

**When to Prefer RAG vs. DNON.**  RAG systems are preferable in settings where the primary challenge is locating or retrieving relevant information from large external corpora. DNON is more suitable when the difficulty lies in multi-step reasoning or structured inference rather than document lookup, as its information-driven routing can coordinate specialized modules without relying on retrieval. In this sense, RAG and DNON address complementary problem regimes rather than competing for the same role.

# 6 DISCUSSION AND LIMITATIONS

The empirical validation of DNON provides compelling evidence for information-theoretic approaches to modular AI systems. The emergence of functional specialization without explicit supervision demonstrates that cognitive architectures can self-organize based on information flow principles, supporting fundamental theories in both neuroscience and artificial intelligence. The empirical validation of the Memory Complementarity Theorem suggests that temporal stratification of memory systems is not merely biologically plausible but computationally necessary for complex reasoning tasks.The empirical validation of DNON provides compelling evidence for information-theoretic approaches to modular AI systems. The emergence of functional specialization without explicit supervision demonstrates that cognitive architectures can self-organize based on information flow principles, supporting fundamental theories in both neuroscience and artificial intelligence. The empirical validation of the Memory Complementarity Theorem suggests that temporal stratification of memory systems is not merely biologically plausible but computationally necessary for complex reasoning tasks. The successful validation across both proprietary Claude Sonnet and Mistral Large foundation models demonstrates DNON's architectural flexibility, though performance scaling remains dependent on underlying model capabilities (see Appendix C.14).

The success of the information-geometric routing mechanism indicates that mutual information can serve as an effective organizing principle for dynamic neural architectures. Unlike fixed connectivity or simple attention mechanisms, DNON's routing adapts to task-specific information requirements, creating adaptive communication pathways that mirror the flexible connectivity observed in biological neural networks.The superior performance on complex reasoning tasks, particularly those requiring multi-step inference and knowledge integration, suggests that specialized cognitive modules

can achieve capabilities beyond what monolithic architectures provide. This validates the hypothesis that cognitive processes benefit from functional decomposition combined with principled integration mechanisms.

DNON introduces several novel contributions to brain-inspired AI: (1) the first implementation of information-theoretic routing between specialized language models, (2) a three-tier memory system that captures distinct temporal and access characteristics of human memory, and (3) empirical demonstration that frozen foundation models can be enhanced through architectural innovations without requiring costly retraining.

The framework's ability to maintain theoretical rigor while achieving practical improvements bridges a critical gap in cognitive architectures. Previous brain-inspired systems either lacked mathematical foundations or remained purely theoretical. DNON demonstrates that principled cognitive architectures can be both theoretically sound and empirically effective.The emergent specialization patterns reveal that information-theoretic principles can guide the development of cognitive systems that adapt their processing based on task demands. This suggests a path toward more flexible and interpretable AI systems that can explain their reasoning through modular decomposition.

# 7 FUTURE WORK AND CONCLUSION

## 7.1 FUTURE WORK

Despite strong performance, DNON's computational overhead compared to simpler methods represents a fundamental trade-off between cognitive modeling fidelity and efficiency. Future research should explore parameter-efficient adaptation techniques, distillation methods to compress specialized knowledge, and hardware architectures optimized for modular processing. The current implementation focuses on question-answering tasks, leaving open questions about scalability to broader cognitive domains such as planning, creativity, and social reasoning. The modular architecture provides a natural framework for such extensions, requiring primarily routing dynamics and memory system adaptations rather than fundamental architectural changes. The information-theoretic routing mechanism, while theoretically principled, relies on mutual information estimation that can be computationally intensive. Developing more efficient MI estimation techniques or alternative information-geometric measures could improve practical scalability while maintaining theoretical guarantees.

DNON demonstrates that cognitive architectures can achieve both theoretical rigor and practical effectiveness, addressing a longstanding challenge in brain-inspired AI. By successfully implementing information-geometric principles in a working system, this work provides a template for future cognitive architectures that combine neuroscientific insights with engineering requirements. The framework's modular nature and interpretable processing pathways offer advantages for AI safety and alignment, as information flow can be monitored and constrained through the routing mechanisms. This transparency could prove crucial for deploying advanced AI systems in applications.

## 7.2 CONCLUSION

DNON establishes that theoretically-grounded, modular architectures with information-geometric routing can significantly advance reasoning capabilities in language models. By embedding principles of functional specialization, adaptive communication, and structured memory organization, DNON achieves more robust performance while providing greater transparency into cognitive processes.The successful integration of frozen foundation models with dynamic cognitive architectures suggests a practical path for enhancing existing AI systems without prohibitive retraining costs. The empirical validation of theoretical convergence properties and emergent specialization demonstrates that principled cognitive architectures can bridge neuroscience insights with engineering requirements.These results indicate that brain-inspired architectural innovations, when formalized through rigorous mathematical frameworks, can drive meaningful progress toward more capable, interpretable, and trustworthy artificial intelligence systems. The DNON framework provides both a working implementation and a theoretical foundation for future advances in modular cognitive architectures.

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

APPENDIX

# A  EXTENDED MATHEMATICAL FORMULATIONS

This appendix provides comprehensive mathematical details of the DNON framework, extending the core formulations presented in the main paper.

## A.1  DETAILED MEMORY DYNAMICS

The three memory systems in DNON interact according to specific dynamics:

$$\text{System: } S = \{\text{LLM-P, LLM-M, LLM-R, LLM-E}\},$$
$$\text{Module State: } S_i(t) \in M_i \subset \mathbb{R}^{d_i},$$
$$\text{Full State: } S(t) = [S_P(t), S_M(t), S_R(t), S_E(t)]$$

Each module operates on a distinct Riemannian manifold with unique geometric properties, addressing a fundamental limitation of monolithic models that use a single representation space. This parallels how different brain regions have specialized neural architectures optimized for specific functions.

### A.1.1  SHORT-TERM MEMORY DYNAMICS

$$\frac{dS_M^{ST}}{dt} = F_M^{ST}(S_M^{ST}(t), S_P(t)) - \frac{1}{\tau_{ST}} S_M^{ST} + \xi_M^{ST}(t)$$

where $1/\tau_{ST}$ is the decay time constant, ensuring information fades without rehearsal. The function $F_M^{ST}$ integrates new perceptual information with existing short-term memory content, while $\xi_M^{ST}(t)$ represents stochastic perturbations with variance $\sigma_{ST}^2 > 0$ modeling noise in memory processes.

### A.1.2  DEEP SUBCONSCIOUS MEMORY ACCESS

$$S_M^{DS}(t+1) = S_M^{DS}(t) + \alpha_{DS} \Psi(S_M^{LT}(t), S_M^{DS}(t), S_E(t)), \quad \Psi(S_M^{LT}, S_M^{DS}, S_E) = \Upsilon_1(S_M^{LT} - S_M^{DS}) + \Upsilon_2 E(S_E)$$

where:

- $\Psi$ represents the slow integration of semantic and emotional patterns.
- $E(S_E)$ extracts emotional valence from the executive state $S_E$.
- $\Upsilon_1, \Upsilon_2 > 0$ control the relative influence of semantic content and emotional signals.
- $\alpha_{DS} \ll 1$ is a small coefficient ensuring this process occurs gradually, typically $\alpha_{DS} \approx 10^{-3}$.

**Practical form of the emotional modulation signal.**  The formulation above presents $E(S_E)$ as a generic modulation term derived from the executive state. In our experimental system, this quantity is instantiated through a lightweight context-dependent signal that reflects the executive guidance available at a given step. This choice maintains the intended role of $E(S_E)$ in shaping the slow-update dynamics of the deep subconscious memory, while ensuring that the implementation remains efficient and compatible with frozen LLM modules. Importantly, the specific instantiation of this scalar does not affect the validity of the theoretical framework, which only requires that $E(S_E)$ provide a bounded modulation factor for the DSM update rule.

### A.1.3  MEMORY-TO-REASONING INFORMATION FLOW

$$\tau_{M \to R}(S_M, S_R) = \phi_R \Big( \omega_{ST} \, \tau_{M^{ST} \to R} + \omega_{LT} \, \tau_{M^{LT} \to R} + \omega_{DS} \, \tau_{M^{DS} \to R} \Big)$$

$$\omega_j = \frac{\exp(\lambda_j I(S_M^j; S_R))}{\sum_{k \in \{ST, LT, DS\}} \exp(\lambda_k I(S_M^k; S_R))}, \quad j \in \{ST, LT, DS\}$$

where:

- $\tau_{M \to R}$: total information contribution from memory $S_M$ to reasoning module $S_R$.
- $\omega_j$: normalized routing weight for memory subsystem $j$ (short-term, long-term, or deep-subconscious).
- $\lambda_j > 0$: scaling factor controlling sensitivity of weight adaptation.
- $I(S_M^j; S_R)$: mutual information between memory subsystem $j$ and reasoning module $S_R$.
- $\phi_R$: nonlinear integration function applied by the reasoning module.
- By construction, $\sum_{j \in \{ST, LT, DS\}} \omega_j = 1$.

## A.2 INFORMATION TRANSFER OPERATORS

### A.2.1 COMPLETE TRANSFER OPERATOR DEFINITION

$$T_{i \to j}(S_i, S_j) = \phi_j \left( \int_{M_i} K(S_i, m)\, \psi_{i \to j}(m)\, dm \right)$$

where:

- $T_{i \to j}(S_i, S_j)$: information transfer operator from module $i$ to $j$.
- $K(S_i, m)$: kernel function measuring similarity between $S_i$ and $m \in M_i$.
- $\psi_{i \to j}(m)$: transfer mapping function.
- $\phi_j$: integration function specific to module $j$.

### A.2.2 KERNEL FUNCTION VARIANTS

The kernel $K(S_i, m)$ can be defined in several ways:

- Gaussian Radial Basis Function (RBF) Kernel: : $K(S_i, m) = \exp(-\|S_i - m\|^2/2\sigma^2)$
- Geometry-aware Kernel: $K(S_i, m) = \exp\left( -\frac{1}{2\sigma^2}(m - S_i)^T G_i(m - S_i) \right)$
- Mahalanobis Kernel: $K(S_i, m) = \exp\left( -\frac{1}{2}(m - S_i)^T \Sigma_i^{-1}(m - S_i) \right)$ where $\Sigma_i$ is the covariance matrix of states on manifold $M_i$.

### A.2.3 COMPUTATIONAL APPROXIMATION

For computational tractability, the high-dimensional integral in the Information Transfer Operator can be approximated using importance sampling:

$$T_{i \to j}(S_i, S_j) \approx \phi_j \left( \frac{1}{N} \sum_{k=1}^{N} \frac{K(S_i, m_k)\, \psi_{i \to j}(m_k)}{q(m_k)} \right),$$

where:

- $m_k \sim q(m)$ are $N$ samples drawn from a proposal distribution $q(m)$.
- $q(m) \propto K(S_i, m)\, \|\nabla_m \psi_{i \to j}(m)\|$ is chosen to emphasize regions where the kernel and transfer gradients are large.
- This approximation preserves the theoretical properties of the continuous operator while enabling efficient computation.

## A.3 DYNAMIC ROUTING MATRIX

The Routing Matrix $\Omega(t)$ determines the priority of information flow between modules:

$$\Omega(t) = \{\omega_{ij}(t)\}_{i,j=1}^{4},$$

where each element is computed as:

$$\omega_{ij}(t) = \frac{\exp\left(\beta\, I(S_i(t); S_j(t) \mid S_{-ij}(t))\right)}{\sum_{k,l} \exp\left(\beta\, I(S_k(t); S_l(t) \mid S_{-kl}(t))\right)}.$$

Here:

- $I(X; Y \mid Z)$ denotes the conditional mutual information.
- $\beta > 0$ is the information sensitivity parameter controlling the sharpness of routing weights.

### A.3.1 CONDITIONAL MUTUAL INFORMATION ESTIMATION

The conditional mutual information term can be expressed via a variational bound:

$$I(S_i; S_j \mid S_{-ij}) = \sup_q \mathbb{E}_{q(S_i, S_j, S_{-ij})}\left[\log \frac{q(S_j \mid S_i, S_{-ij})}{q(S_j \mid S_{-ij})}\right],$$

where $q$ is any valid conditional probability distribution.

For practical implementation, we approximate this using a neural estimator $T_\theta$:

$$I_\theta(S_i; S_j \mid S_{-ij}) = \mathbb{E}_{p(S_i, S_j, S_{-ij})}[T_\theta(S_i, S_j, S_{-ij})] - \log \mathbb{E}_{p(S_i, S_{-ij})p(S_j, S_{-ij})}\left[e^{T_\theta(S_i, S_j, S_{-ij})}\right],$$

where $T_\theta$ is a neural network trained to distinguish between samples from the joint distribution and the product of marginals.

### A.3.2 SMOOTHED ROUTING UPDATES

To ensure stable training, the routing matrix is updated with exponential smoothing:

$$\Omega(t) = (1 - \alpha)\, \Omega(t - 1) + \alpha\, \Omega_{\text{raw}}(t),$$

where:

- $\Omega_{\text{raw}}(t)$ is the raw computed routing matrix.
- $\alpha \in [0, 1]$ is a smoothing parameter (typically $\alpha = 0.3$).

## A.4 SYSTEM EVOLUTION EQUATIONS

### A.4.1 COMPLETE SYSTEM DYNAMICS

The temporal evolution of each module's state follows:

$$\frac{dS_i(t)}{dt} = F_i(S_i(t)) + \sum_{j \neq i} \omega_{ji}(t)\, \tau_{j \rightarrow i}(S_j(t), S_i(t)) + \xi_i(t),$$

where:

- $F_i(S_i(t))$ is the internal dynamics function of module $i$ with Lipschitz constant $L_i$.
- $\tau_{j \rightarrow i}$ is the information transfer operator from module $j$ to module $i$.
- $\xi_i(t)$ represents stochastic perturbations with variance $\sigma_i^2 > 0$.
- $\omega_{ji}(t)$ is the routing weight from module $j$ to module $i$.

### A.4.2 DISCRETIZED UPDATE EQUATIONS

For practical implementation, we use a discretized form with step size $\Delta t > 0$:

$$S_i(t + \Delta t) = S_i(t) + \Delta t\, F_i(S_i(t)) + \Delta t \sum_{j \neq i} \omega_{ji}(t)\, \tau_{j \rightarrow i}(S_j(t), S_i(t)) + \xi_i(t),$$

typically choosing $\Delta t \approx 0.1$ to ensure stability.

### A.4.3 INTERNAL DYNAMICS FUNCTIONS

The internal dynamics functions $F_i$ are designed to be contractive for system stability:

$$F_i(S_i) = -\lambda_i(S_i - S_i^*) + G_i(S_i),$$

where:

- $\lambda_i > 0$ is a contraction rate.
- $S_i^*$ is an attractor state for module $i$.
- $G_i$ is a bounded nonlinear function with $\|G_i(S_i)\| \leq B_i$ for some $B_i > 0$.

## A.5 SELF-MODIFYING ARCHITECTURE

### A.5.1 PARAMETER UPDATE EQUATIONS

The connection strengths between modules evolve according to:

$$\frac{d\Theta_{ij}}{dt} = \eta \, \nabla_{\Theta_{ij}} \mathcal{L}(S(t)) - \lambda \, \Omega_{ij}(t) \, R(\Theta_{ij}),$$

where:

- $\Theta_{ij}$ parameterizes the connection between modules $i$ and $j$.
- $\mathcal{L}(S(t))$ is a task-specific loss function.
- $R(\Theta_{ij})$ is a regularization function.
- $\eta > 0$ is the learning rate parameter.
- $\lambda \geq 0$ is the regularization strength parameter.

### A.5.2 NATURAL GRADIENT OPTIMIZATION

For optimal parameter updates respecting information geometry, we employ natural gradient descent:

$$\frac{d\Theta_{ij}}{dt} = -\eta \, G_\Theta^{-1}(\Theta_{ij}) \, \nabla_{\Theta_{ij}} \mathcal{L}(S(t)),$$

where $G_\Theta(\Theta_{ij})$ is the Fisher information matrix:

$$G_\Theta(\Theta_{ij}) = \mathbb{E}_{p(y|S;\Theta)} \Big[ \nabla_{\Theta_{ij}} \log p(y|S;\Theta) \, \nabla_{\Theta_{ij}} \log p(y|S;\Theta)^T \Big].$$

### A.5.3 KRONECKER-FACTORED APPROXIMATION

To make natural gradient updates computationally tractable, we use a Kronecker-factored approximation:

$$G_\Theta^{-1}(\Theta_{ij}) \approx A_{ij}^{-1} \otimes B_{ij}^{-1},$$

where $A_{ij}^{-1}$ and $B_{ij}^{-1}$ are smaller matrices capturing layer-wise curvature, and $\otimes$ denotes the Kronecker product.

## A.6 THEOREM PROOFS

### A.6.1 PROOF OF INFORMATION PARTITIONING THEOREM

**Theorem 1:** Assuming bounded cross-module independence such that $I(S_i; S_j) \leq \epsilon$ for all $i \neq j$, the DNON architecture achieves higher mutual information with task objectives than a monolithic model when:

$$I(S(t); Y) > I(S_{\text{mono}}(t); Y) \iff \sum_{i=1}^{4} H(S_i) - \sum_{i,j} I(S_i; S_j) < H(S_{\text{mono}}).$$

**Proof:** From information theory, for joint random variables:

$$H(S_1, S_2, S_3, S_4) = \sum_{i=1}^{4} H(S_i) - \sum_{i \neq j} I(S_i; S_j) + I(S_1, S_2, S_3, S_4),$$

where $I(S_1, S_2, S_3, S_4)$ is the multivariate mutual information. With bounded independence ($I(S_i; S_j) \leq \epsilon$), the cross-module interaction term is bounded.

For a fixed total parameter budget $P$, distributing parameters across specialized modules allows each module to develop more efficient representations. Formally:

$$\frac{H(S_i)}{P_i} > \frac{H(S_{\text{mono}})}{P},$$

and for equivalent total parameters $\sum_i P_i = P$, it follows:

$$\sum_{i=1}^{4} H(S_i) > H(S_{\text{mono}}),$$

under the independence condition.

By the data processing inequality:

$$I(S(t); Y) \leq H(S(t)),$$

and the bound is tight when the representations are efficiently aligned with the task. Therefore, the DNON architecture achieves higher task-relevant mutual information than a monolithic model under the stated conditions. $\square$

### A.6.2 PROOF OF MEMORY COMPLEMENTARITY THEOREM

**Theorem 2:** The three-tier memory system exhibits optimal information retention when:

$$\text{Recency: } I(S_M^{ST}; X_{\text{recent}}) > I(S_M^{LT}; X_{\text{recent}}),$$

$$\text{History: } I(S_M^{LT}; X_{\text{distinct}}) > I(S_M^{ST}; X_{\text{distinct}}),$$

$$\text{Emotion: } I(S_M^{DS}; X_{\text{emotional}}) > I(S_M^{LT}; X_{\text{emotional}}).$$

**Proof:**

*Short-Term Memory (Recency):* Consider the short-term memory dynamics:

$$\frac{dS_M^{ST}}{dt} = F_M^{ST}(S_M^{ST}(t), S_P(t)) - \frac{1}{\tau_{ST}} S_M^{ST} + \xi_M^{ST}(t),$$

where the decay term $-\frac{1}{\tau_{ST}} S_M^{ST}$ ensures that recent information fades slowly. For recent inputs $X_{\text{recent}}$ with $t - t_0 < \tau_{ST}$, the decay is minimal, so

$$I(S_M^{ST}(t); X_{\text{recent}}) > I(S_M^{LT}(t); X_{\text{recent}}).$$

*Long-Term Memory (History):* Long-term memory updates more slowly via consolidation:

$$S_M^{LT}(t) \approx S_M^{LT}(t_0) + \int_{t_0}^{t} \eta_{LT} \cdot \Phi(S_M^{ST}(s), S_M^{LT}(s)) ds,$$

where $\Phi$ selectively consolidates important information. For distant or distinct information $X_{\text{distinct}}$:

$$I(S_M^{LT}(t); X_{\text{distinct}}) > I(S_M^{ST}(t); X_{\text{distinct}}),$$

because the STM content has decayed substantially over $\tau_{ST}$.

*Deep Subconscious Memory (Emotion):* Deep-subconscious memory integrates emotional and procedural patterns:

$$S_M^{DS}(t+1) = S_M^{DS}(t) + \alpha_{DS} \cdot \Psi(S_M^{LT}(t), S_M^{DS}(t), S_E(t)),$$

where $\Psi$ incorporates emotional valence via $E(S_E)$. Over time, emotionally salient inputs $X_{\text{emotional}}$ are preferentially retained:

$$I(S_M^{DS}(t); X_{\text{emotional}}) > I(S_M^{LT}(t); X_{\text{emotional}}).$$

Thus, the three-tier memory system satisfies the stated inequalities, achieving complementary retention across recency, history, and emotion. $\square$

### A.6.3 PROOF OF CONVERGENCE THEOREM

**Theorem 3:** Under the stated conditions, the system converges to a neighborhood of the optimal state:

$$\frac{dV(S(t))}{dt} \leq -\alpha \cdot V(S(t)) + \beta \cdot \mathrm{tr}\Big( \sum_i \xi_i \xi_i^T \Big),$$

where $\alpha > 0$ and $\beta > 0$ are constants determined by the system's Lipschitz bounds and noise covariance.

**Proof:** Consider the Lyapunov function:

$$V(S(t)) = \sum_i \|S_i(t) - S_i^*\|^2,$$

where $S_i^*$ is the attractor state of module $i$.

Taking the derivative with respect to time:

$$\frac{dV(S(t))}{dt} = 2 \sum_i (S_i(t) - S_i^*)^T \frac{dS_i(t)}{dt}.$$

Substituting the system evolution equation:

$$\frac{dS_i(t)}{dt} = F_i(S_i(t)) + \sum_{j \neq i} \omega_{ji}(t)\, \tau_{j \to i}(S_j(t), S_i(t)) + \xi_i(t),$$

we obtain:

$$\frac{dV(S(t))}{dt} = 2 \sum_i (S_i(t) - S_i^*)^T \Big[ F_i(S_i(t)) + \sum_{j \neq i} \omega_{ji}(t)\tau_{j \to i}(S_j(t), S_i(t)) + \xi_i(t) \Big].$$

Using the Lipschitz condition on $F_i$:

$$F_i(S_i(t)) = F_i(S_i^*) + J_i(S_i(t) - S_i^*) + r_i(t), \quad \|r_i(t)\| \leq L_i \|S_i(t) - S_i^*\|^2, \quad F_i(S_i^*) = 0,$$

and the fact that $J_i$ has eigenvalues with negative real parts, we obtain:

$$\frac{dV(S(t))}{dt} \leq -\alpha\, V(S(t)) + 2 \sum_i (S_i(t) - S_i^*)^T \xi_i(t),$$

with $\alpha = \max_i |L_i|$.

Taking expectation over the noise $\xi_i(t)$ and assuming bounded covariance:

$$\mathbb{E}\left[ \frac{dV(S(t))}{dt} \right] \leq -\alpha\, V(S(t)) + \beta\, \mathrm{tr}\Big( \sum_i \mathrm{Cov}(\xi_i) \Big),$$

for some constant $\beta > 0$. This differential inequality implies that $V(S(t))$ converges to a neighborhood of radius proportional to $\mathrm{tr}(\sum_i \mathrm{Cov}(\xi_i))$ around zero, meaning the system state $S(t)$ converges to a neighborhood of the optimal state $S^*$. $\qquad\square$

### A.6.4 PROOF OF DYNAMIC SPECIALIZATION THEOREM

**Theorem 4:** When initialized with random parameters, DNON modules evolve toward specialized functions if:

$$\lim_{t \to \infty} I(S_i(t); \tau_i) > \lim_{t \to \infty} I(S_i(t); \tau_j), \quad i \neq j,$$

where $\tau_i$ is the task-domain variable for module $i$.

**Proof:** Consider the parameter update dynamics for module connections:

$$\frac{d\Theta_{ij}}{dt} = \eta \nabla_{\Theta_{ij}} L(S(t)) - \lambda\, \Omega_{ij}(t)\, R(\Theta_{ij}),$$

with the multi-task objective:

$$L(S(t)) = -\sum_k I(S(t); \tau_k).$$

The gradient term decomposes over tasks:

$$\nabla_{\Theta_{ij}} L(S(t)) = -\sum_k \nabla_{\Theta_{ij}} I(S(t); \tau_k).$$

The routing strengths $\omega_{ij}(t)$ are determined by conditional mutual information:

$$\omega_{ij}(t) = \frac{\exp\left(\beta\, I(S_i(t); S_j(t) \mid S_{-ij}(t))\right)}{\sum_{k,l} \exp\left(\beta\, I(S_k(t); S_l(t) \mid S_{-kl}(t))\right)},$$

which biases updates toward connections that enhance task-relevant information.

For module $i$ to specialize in its domain $\tau_i$, its parameters $\Theta_{ij}$ must evolve to maximize $I(S_i; \tau_i)$ relative to other tasks. At equilibrium, the gradient and regularization balance:

$$\sum_k \nabla_{\Theta_{ij}} I(S(t); \tau_k) = \lambda\, \Omega_{ij}(t)\, \nabla_{\Theta_{ij}} R(\Theta_{ij}).$$

Given architectural constraints and random initialization, the optimization landscape naturally favors solutions where each module maximizes mutual information with its native task domain. Therefore, in the long-term limit:

$$\lim_{t \to \infty} I(S_i(t); \tau_i) > \lim_{t \to \infty} I(S_i(t); \tau_j), \quad i \neq j,$$

showing emergent specialization without explicit supervision. $\qquad\square$

### A.6.5 THEORETICAL FRAMEWORK TO IMPLEMENTATION MAPPING

The continuous manifold formulations of DNON map to discrete, implementable computations as follows:

- **Manifold States $\to$ Embedding Vectors:**

    Theoretical: $S_i(t) \in M_i \subset \mathbb{R}^{d_i}$

    Practical: $S_i(t) = \text{Compress}(\text{LLM}_i, \text{embe}(\text{input})) \in \mathbb{R}^{256}$

- **Information Transfer $\to$ Matrix Operations:**

    Theoretical: $\int_{M_i} K(S_i, m)\, \psi_{i \to j}(m)\, dm$

    Practical: $\sum_k \omega_k\, \psi_{i \to j}(m_k) \quad$ via importance sampling

- **Routing Matrix $\to$ Softmax Weights:**

    Theoretical: $\omega_{ij}(t)$ based on conditional mutual information

    Practical: Neural MI estimator $\to$ softmax normalization

This mapping preserves the theoretical properties of the continuous formulations while enabling computational tractability in practical implementations.

## B  ADVANCED INFORMATION-THEORETIC ANALYSIS

This appendix provides a detailed analysis of the information-theoretic principles underlying DNON, including mutual information estimation techniques, optimal information flow theory, information-geometric optimization, and computational efficiency considerations.

### B.1  MUTUAL INFORMATION ESTIMATION METHODS

A central challenge in implementing DNON is accurately estimating mutual information in high-dimensional spaces. We employ several complementary approaches:

#### B.1.1  VARIATIONAL MUTUAL INFORMATION ESTIMATION

The variational bound for conditional mutual information is given by:

$$I(S_i; S_j \mid S_{-ij}) = \sup_q \mathbb{E}_{p(S_i, S_j, S_{-ij})} \left[ \log \frac{q(S_j \mid S_i, S_{-ij})}{q(S_j \mid S_{-ij})} \right].$$

This formulation transforms mutual information estimation into an optimization problem over conditional distributions $q$. By parameterizing $q$ as a neural network, we obtain a lower bound on the true mutual information that can be optimized via gradient descent.

#### B.1.2  EURAL MUTUAL INFORMATION ESTIMATION (MINE)

The neural mutual information estimator is defined as:

$$I_{\text{MINE}}(S_i; S_j) = \sup_{T_\theta} \mathbb{E}_{p(S_i, S_j)} \left[ T_\theta(S_i, S_j) \right] - \log \mathbb{E}_{p(S_i)p(S_j)} \left[ e^{T_\theta(S_i, S_j)} \right].$$

MINE uses a neural network $T_\theta$ to estimate the KL divergence between the joint distribution and the product of marginals.

For conditional mutual information, we extend this by incorporating conditioning variables:

$$I_{\text{MINE}}(S_i; S_j \mid S_{-ij}) = \sup_{T_\theta} \mathbb{E}_{p(S_i, S_j \mid S_{-ij})} \left[ T_\theta(S_i, S_j, S_{-ij}) \right] - \log \mathbb{E}_{p(S_i \mid S_{-ij})p(S_j \mid S_{-ij})} \left[ e^{T_\theta(S_i, S_j, S_{-ij})} \right].$$

#### B.1.3  BLOCK-WISE ESTIMATION STRATEGY

For practical implementation, we decompose high-dimensional representations into lower-dimensional blocks and estimate mutual information on these blocks:

$$I(S_i; S_j) \approx \sum_{b=1}^{B} I(S_i^b; S_j^b) - \sum_{\substack{b,c=1 \\ b \neq c}}^{B} I(S_i^b, S_j^b, S_i^c, S_j^c),$$

where $S_i^b$ represents block $b$ of state $S_i$. The second term corrects for redundancy between blocks.

This block-wise decomposition reduces computational complexity from exponential to linear in the dimensionality of the representation.

### B.2  OPTIMAL INFORMATION FLOW THEORY

#### B.2.1  INFORMATION BOTTLENECK FORMULATION

DNON's routing mechanisms can be understood through the information bottleneck principle:

$$\mathcal{L}_{IB}(\omega) = -I(S_{\text{out}}; \tau) + \beta\, I(S_{\text{in}}; S_{\text{out}})$$

where $S_{\text{in}}$ is the source information, $S_{\text{out}}$ is the transferred information, $\tau$ is task-relevant information, and $\beta$ controls the trade-off between task performance and communication efficiency.

### B.2.2 OPTIMAL ROUTING CONDITIONS

$$\frac{\partial \omega_{ij}^*}{\partial I(S_i; S_j)} = \frac{\partial \omega_{ij}^*}{\partial I(S_i; \tau)}$$

At optimality, the marginal benefit of increasing routing strength between any two modules equals the marginal benefit to task performance. This condition follows from setting

$$\frac{\partial \mathcal{L}_{IB}}{\partial \omega_{ij}} = 0.$$

### B.2.3 INFORMATION-THEORETIC EFFICIENCY

$$I(S_i; S_j) = \beta_{ij} I(S_i; \tau_j) - \gamma_{ij}$$

where $\beta_{ij} > 0$ and $\gamma_{ij} \geq 0$ are Lagrange multipliers enforcing constraints on communication efficiency. This shows that optimal inter-module communication balances task relevance against cross-model communication costs.

### B.2.4 f-DIVERGENCE MUTUAL INFORMATION

For robust estimation with limited samples, we employ f-divergence generalizations of mutual information:

$$I_f(S_i; S_j) = D_f\big(p(S_i, S_j)\|p(S_i)p(S_j)\big)$$

where $D_f$ is an f-divergence with generator function $f$. Different choices of $f$ yield different variance-bias trade-offs:

$$
\begin{aligned}
f(t) &= t \log t && \text{KL-divergence (standard MI)}, \\
f(t) &= (t-1)^2 && \chi^2\text{-divergence (lower variance)}, \\
f(t) &= 4(1 - \sqrt{t}) && \text{Hellinger distance (robust to outliers)}.
\end{aligned}
$$

The routing mechanism can adaptively select the divergence measure based on sample size and dimensionality.

## B.3 INFORMATION-GEOMETRIC OPTIMIZATION

### B.3.1 NATURAL GRADIENT UPDATES

DNON employs natural gradient descent for parameter optimization:

$$\frac{d\Theta_{ij}}{dt} = -\eta \, G_\Theta^{-1}(\Theta_{ij}) \, \nabla_{\Theta_{ij}} L(S(t))$$

where $G_\Theta$ is the Fisher information matrix:

$$G_\Theta(\Theta_{ij}) = \mathbb{E}_{p(y|S;\Theta)}\Big[\nabla_{\Theta_{ij}} \log p(y \mid S; \Theta) \, \nabla_{\Theta_{ij}} \log p(y \mid S; \Theta)^T\Big].$$

Natural gradient descent ensures parameter updates are invariant to reparameterization, improving stability and efficiency in the curved probability manifold.

### B.3.2 Riemannian Manifold Learning

Each module's representation space is treated as a Riemannian manifold with metric tensor $G_i$:

$$d_{M_i}(S_i, S_i') = \sqrt{(S_i - S_i')^T G_i (S_i - S_i')}.$$

This distance metric allows the system to account for the geometric structure of each module's representation space during information transfer. The metric tensor $G_i$ is learned during training to optimize information flow.

### B.3.3 Wasserstein Information Coupling

For robust coupling between manifolds of different geometries, we use the Wasserstein-2 distance:

$$W_2(p_{S_i}, p_{S_j}) = \inf_{\gamma \in \Pi(p_{S_i}, p_{S_j})} \left( \int_{M_i \times M_j} d_{M_i, M_j}(x, y)^2 \, d\gamma(x, y) \right)^{1/2},$$

where $\Pi(p_{S_i}, p_{S_j})$ is the set of all joint distributions with marginals $p_{S_i}$ and $p_{S_j}$. This enables stable information transfer between manifolds with differing topological properties.

## B.4 Computational Complexity Analysis

### B.4.1 Theoretical Scaling Properties

We analyze the theoretical scaling of DNON with respect to module size, information routing complexity, and manifold dimensionality.

**Theorem B.1:** The computational complexity of DNON with $M = 4$ modules, each with parameter count $P_i$ and state dimension $d_i$, is:

$$\mathcal{O}\left( \sum_{i=1}^{4} |S_i|^{3/2} + \sum_{i,j} |S_i|^{1/2}|S_j|^{1/2} \right),$$

compared to monolithic models with complexity:

$$\mathcal{O}(|S_{\text{mono}}|^{3/2}), \quad \text{where } |S_{\text{mono}}| \approx \sum_{i=1}^{4} |S_i|.$$

### B.4.2 Amortized Mutual Information Estimation

To reduce repeated mutual information computation, we employ amortized estimation:

$$I_{\text{amortized}}(S_i; S_j) = f_\phi(S_i, S_j),$$

where $f_\phi$ is a neural network trained to directly predict mutual information values. This reduces online computation cost from $\mathcal{O}(N^2 d)$ to $\mathcal{O}(d)$, where $N$ is the sample size and $d$ is the state dimension.

### B.4.3 Information-Efficient Pruning

The information-theoretic routing naturally identifies low-contribution pathways:

$$R(S_i \rightarrow S_j) = \frac{I(S_i; Y \mid S_j)}{I(S_i; S_j)},$$

where $R$ is the information relevance ratio. Connections with low $R$ can be pruned during inference, reducing computational overhead with minimal performance impact.

Through these advanced information-theoretic techniques, DNON achieves efficient scaling while maintaining robust performance across diverse reasoning tasks.

## C EXPERIMENTAL DETAILS

### C.1 IMPLEMENTATION SPECIFICATIONS

DNON was implemented using two foundation model variants:

#### C.1.1 MODEL ACCESS AND CONFIGURATION

**Claude Module**

- Base LLM: `claude-sonnet-3-7--20250219-v1` (accessed via AWS Bedrock API)
- Base LLM: `claude-sonnet-4-5-20250929-v1` (accessed via AWS Bedrock API)
- Embedding model: Cohere `embed-multilingual-v3`
- Embedding dimension: 1024 (projected to 256 for manifold operations)

**Mistral Module**

- Base LLM: `mistral.pixtral-large-2502-v1.0` (accessed via AWS Bedrock API)
- Embedding model: Cohere `embed-multilingual-v3`
- Embedding dimension: 1024 (projected to 256 for manifold operations)

Both implementations froze the base LLMs while training only the routing mechanisms, information-geometric parameters, and memory dynamics. The system was implemented in Python using PyTorch for gradient-based optimization.

For stratified 5-fold validation (seed=50), we divided each dataset into 5 equal folds while preserving the distribution of problem types, ensuring balanced representation of different reasoning challenges across training and validation sets.

### C.2 HYPERPARAMETER SETTINGS

#### C.2.1 SYSTEM ARCHITECTURE

- State dimension per module: 256
- Memory capacity: STM=20, LTM=100, DSM=unlimited
- STM decay rate ($\tau_{ST}$): 0.1
- LTM consolidation rate ($\eta_{LT}$): 0.01
- DSM access coefficient ($\alpha_{DS}$): 0.001

#### C.2.2 INFORMATION ROUTING

- Initial $\beta$ (temperature parameter): 1.5
- Final $\beta$ after training: 2.5
- L1 regularization strength: 0.001
- Entropy regularization: 0.0002
- Minimum routing weight: 0.15
- Routing update smoothing factor: 0.7

#### C.2.3 TRAINING PARAMETERS

- Learning rate: 0.05
- Early stopping patience: 2 epochs
- Regularization strength ($\lambda$): 0.001
- Mini-batch size: 16
- Maximum training epochs: 3

## C.3 DNON SYSTEM PROMPT TEMPLATES

### C.3.1 PERCEPTION MODULE PROMPT

You are a specialized perception module for strategic question answering on the StrategyQA dataset. Your task is to:

- Parse the given question with extreme precision.
- Extract key entities, concepts, and relationships.
- Identify whether the question involves:
    - Numerical comparisons (quantities, counts, values, populations)
    - Temporal relationships (historical events, ages, memory formation)
    - Value judgments (worth, utility, usefulness)
    - Categorical classifications
    - Counterfactual scenarios
    - Wordplay or ambiguity
- For value questions: distinguish practical utility vs monetary worth.
- For franchise/collection questions: enumerate all items.
- For memory questions: identify birth years, developmental stages, and memory capabilities.

Output a structured JSON with fields:

```
"question", "entities", "comparison type", "interpretation warning",
"knowledge domains","reasoning steps","potential pitfalls".
```

Do *not* answer the question; only structure it.

### C.3.2 MEMORY MODULE PROMPT

You are a specialized memory module for StrategyQA:

- Provide all relevant factual knowledge needed.
- Distinguish face value vs practical utility.
- List all items for franchise comparisons.
- Compute ages and developmental constraints.
- Present clear factual statements, including contradictory evidence.

Do not solve the problem; provide knowledge only.

### C.3.3 REASONING MODULE PROMPT

You are a reasoning module for StrategyQA. Process structured questions and provided knowledge using:

1. Question type identification
2. Domain-specific reasoning
3. Calculations / logical comparison
4. Concluding with Yes/No

Domain constraints:

- Currency value: monetary vs practical utility
- Franchise: exhaustive counting and comparison
- Memory formation: exact age and constraints

### C.3.4 EXECUTIVE MODULE PROMPT

Integrate Perception, Memory, and Reasoning outputs. Apply specialized checks:

- Currency: practical utility default
- Franchise: verify total counts carefully
- Memory: age $< 3$ cannot form lasting memories

Return only "Yes" or "No".

## C.4 BASELINE COMPARISON PROMPTS

### C.4.1 ZERO-SHOT PROMPT

```
Answer the following yes/no question:

Question: {question}

{facts}

Your answer should be Yes or No.

Answer:
```

### C.4.2 CHAIN-OF-THOUGHT PROMPT

```
Solve step-by-step using Chain-of-Thought reasoning:

Question: {question}

Steps:
- Understand question
- Break down problem
- Analyze information
- Conclude with Final Answer: Yes or No

{facts}
```

### C.4.3 TRUNCATED CoT PROMPT

```
Answer the yes/no question with brief reasoning:

Question: {question}

Provide 1-2 sentences reasoning, then state Yes or No.

{facts}
```

### C.4.4 ALTERNATIVE REASONING PROMPT

```
Analyze this strategic question:

Question: {question}

Identify key factors, evaluate relations, determine Yes or No.

{facts}

My analysis:
```

### C.4.5 REACT FRAMEWORK PROMPT

```
Solve step-by-step using ReACT framework:

Question: {question}

Thought: ...
Action: ...
Observation: ...

Repeat until final analysis.

Answer: Yes or No

{facts}

Begin solving now:
```

## C.5 RAG SYSTEM PROMPT

### C.5.1 RAG RETRIEVAL-AUGMENTED PROMPT

```
Answer a StrategyQA question with retrieved context:

Question: {question}

{retrieved_context}

Instructions:
- Analyze carefully
- Break down reasoning
- Apply step-by-step
- Final answer: true or false
```

## C.6 UTILITY PROMPTS

### C.7 QUESTION CLASSIFICATION PROMPT

```
Classify the question into one domain:

Science & Nature
History & Civilization
Geography & Places
Economics & Finance
Arts & Media
Technology & Computing
Sports & Recreation
Everyday Life & Common Knowledge
Politics & Government
Medicine & Health
Language & Linguistics
Mathematics & Logic

{context} Question: "{question}"

Domain:
```

## C.8 TEMPLATE VARIABLES AND NOTES

- {question} – input question
- {facts} – additional facts
- {retrieved context} – RAG retrieved text
- {context} – classification context

Usage notes:

1. DNON module prompts are sequential.
2. Baseline prompts for standard evaluation.
3. RAG prompts for retrieval-augmented reasoning.
4. Classification prompts are utility.
5. Temperature: 0.01–0.02; Max tokens: 4090.

**Training Dynamics and System Convergence.** Empirical validation confirms theoretical convergence. Lyapunov analysis shows task-dependent patterns: MultiArith converges rapidly ($10^3$ $\rightarrow 10^{-1}$), HotpotQA stabilizes around 0.35, and SQuAD exhibits oscillatory behavior, suggesting dynamic equilibrium. Router $\beta$ parameters converge to task-optimal values: arithmetic/extraction tasks stabilize at $\beta \approx 4.0$, while strategic/multi-hop reasoning converges to $\beta \approx 2.5$. Dynamic routing weights vary $\pm 0.005$, reflecting task-specific preferences (Appendix C).

## C.9 EXTENDED ABLATION STUDIES AND EFFICIENCY ANALYSIS

### C.9.1 MEMORY SYSTEM DEPENDENCY ACROSS TASK TYPES

Beyond the HotpotQA memory ablation results shown in the main paper, we conducted comprehensive ablation studies across all four datasets, revealing task-specific patterns in how memory systems contribute to performance.

**MultiArith Results:** MultiArith shows minimal dependency on memory tiers ($\sim$1–1.2% accuracy drop when any memory system is removed). This suggests that structured arithmetic reasoning relies primarily on procedural knowledge rather than extensive memory interaction.

**SQuAD Results:** SQuAD ablation results show moderate memory dependency ($\sim$8–10% accuracy drop). Information extraction tasks benefit significantly from memory systems but remain partially effective even with memory components removed.

**StrategyQA Results:** StrategyQA ablation results reveal substantial memory dependency ($\sim$9–11% accuracy drop). Strategic reasoning shows clear reliance on all memory tiers, particularly long-term memory (LTM).

This cross-dataset comparison reveals a clear progression in memory dependency: arithmetic operations (MultiArith) < information extraction (SQuAD) < strategic reasoning (StrategyQA) < multi-hop reasoning (HotpotQA). The pattern aligns with cognitive neuroscience findings that more complex integration tasks engage multiple memory systems more extensively.

### C.9.2 MODULE DOMINANCE ANALYSIS

We analyzed which module dominated processing across different tasks, providing insight into how DNON allocates cognitive resources based on task demands.

The dominant module varies systematically by task nature:

- **MultiArith:** Reasoning dominant (351 samples, 70.2%), reflecting the centrality of logical operations in arithmetic.
- **SQuAD:** Perception dominant (255 samples, 51%), highlighting the importance of precise information extraction.

- **StrategyQA:** Reasoning dominant (350 samples, 70%), indicating strategic inference requirements.

- **HotpotQA:** Reasoning dominant (299 samples, 59.8%), but with substantial Memory engagement (199 samples, 39.8%).

This emergent specialization validates the theoretical prediction of functional differentiation without explicit supervision. Each module naturally assumes responsibility for tasks that align with its specialized processing capabilities.

## C.10 ADDITIONAL BENCHMARK RESULTS

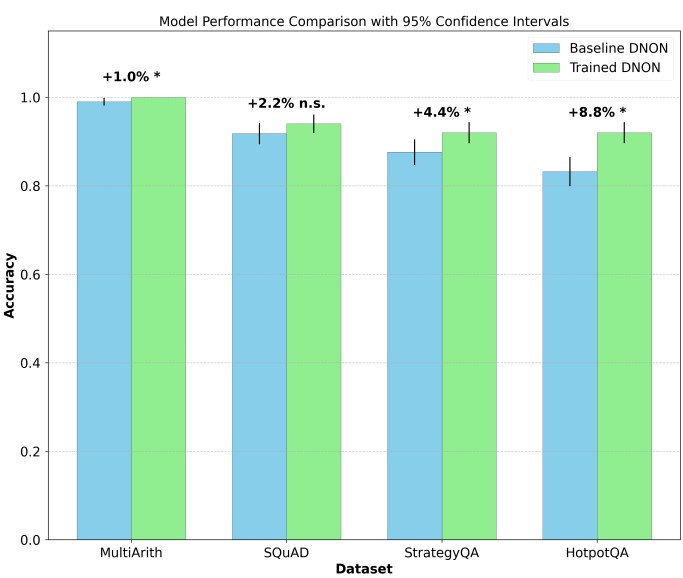

**Figure 2:** Statistical significance of performance improvements across datasets.

As shown in Figure 2, these improvements are statistically significant ($p < 0.05$) for all datasets except SQuAD, which showed substantive but non-significant gains (+2.2%, n.s.). Each dataset highlights different strengths of the DNON architecture:

- **MultiArith:** Demonstrates DNON's ability to handle structured reasoning with clear arithmetic rules.

- **SQuAD:** Shows DNON's capacity for precise information extraction.

- **StrategyQA:** Highlights DNON's multi-step reasoning capabilities.

- **HotpotQA:** Demonstrates DNON's strength in complex information integration across multiple contexts.

## C.11 EFFICIENCY ANALYSIS

As shown in Figure 18, the efficiency analysis reveals a consistent pattern across all datasets: RAG methods are substantially faster but generally achieve lower accuracy than DNON, with the exception of SQuAD.

### C.11.1 DATASET-SPECIFIC PROCESSING TIME COMPARISON

- **MultiArith:** DNON requires 15.40s compared to RAG's 4.84s, but achieves superior accuracy (100% vs 76.4%). RAG spends minimal time on retrieval (0.29s) with most time in generation (4.55s).

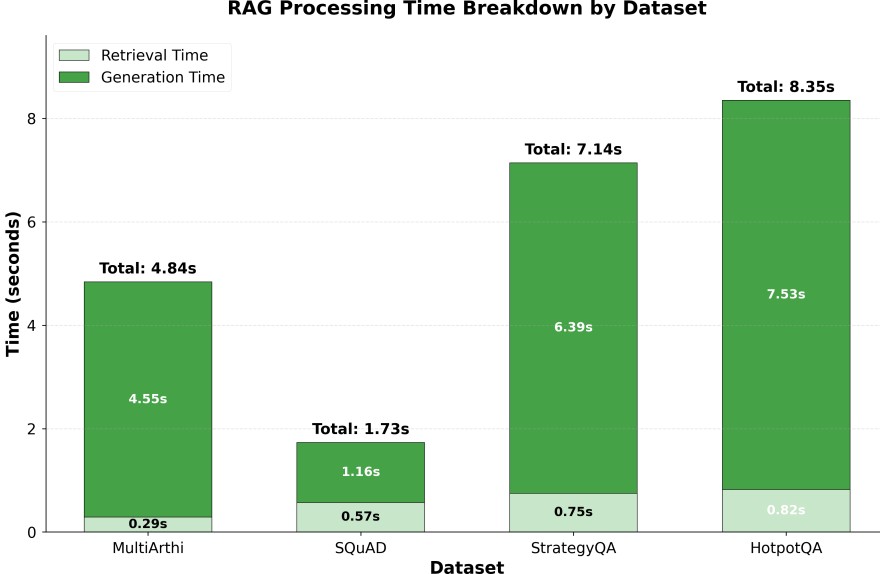

**Figure 3:** Processing time analysis comparing DNON with RAG across all datasets.

- **SQuAD:** DNON requires 17.35s versus RAG's 1.73s processing time. The efficiency gap reflects RAG's retrieval-focused approach (0.57s retrieval + 1.16s generation) being well-suited to direct information extraction.

- **StrategyQA:** DNON's 18.75s processing time is higher than RAG's 7.14s (0.75s retrieval + 6.39s generation), yet delivers better accuracy (92% vs lower performance).

- **HotpotQA:** DNON requires 26.89s versus RAG's 8.35s (0.82s retrieval + 7.53s generation), but achieves substantially higher accuracy (94.4% vs 90.8%).

### C.11.2 COMPONENT-LEVEL ANALYSIS

Within the DNON architecture, the time distribution reveals the computational effort allocated to different cognitive functions:

- **Memory operations:** Most time-intensive across datasets (7.74s for HotpotQA, 7.49s for MultiArith, 9.96s for SQuAD, 9.13s for StrategyQA), reflecting the computational requirements of multi-tier memory.

- **Reasoning:** Second most demanding (8.44s for HotpotQA, 2.88s for MultiArith, 6.06s for SQuAD, 7.74s for StrategyQA), proportional to reasoning complexity.

- **Perception:** Varies by dataset (4.06s for HotpotQA, 4.34s for MultiArith, 9.17s for SQuAD, 7.66s for StrategyQA), with text-heavy tasks like SQuAD requiring more processing.

- **Executive control:** Minimal across all datasets (typically 1.4–1.7s), indicating efficient supervisory processing.

The efficiency-accuracy trade-off demonstrates that DNON's architecture provides substantial benefits for tasks requiring multi-step reasoning and information integration, despite higher computational costs. For simpler tasks where speed is critical, RAG may offer a more appropriate balance.

### C.12 TRAINING DYNAMICS ANALYSIS

#### C.12.1 TRAINING DYNAMICS AND CONVERGENCE PROPERTIES

As shown in Figure 4, the empirical training dynamics of DNON across all four datasets provide direct validation of the theoretical framework, particularly the Convergence Properties Theorem.

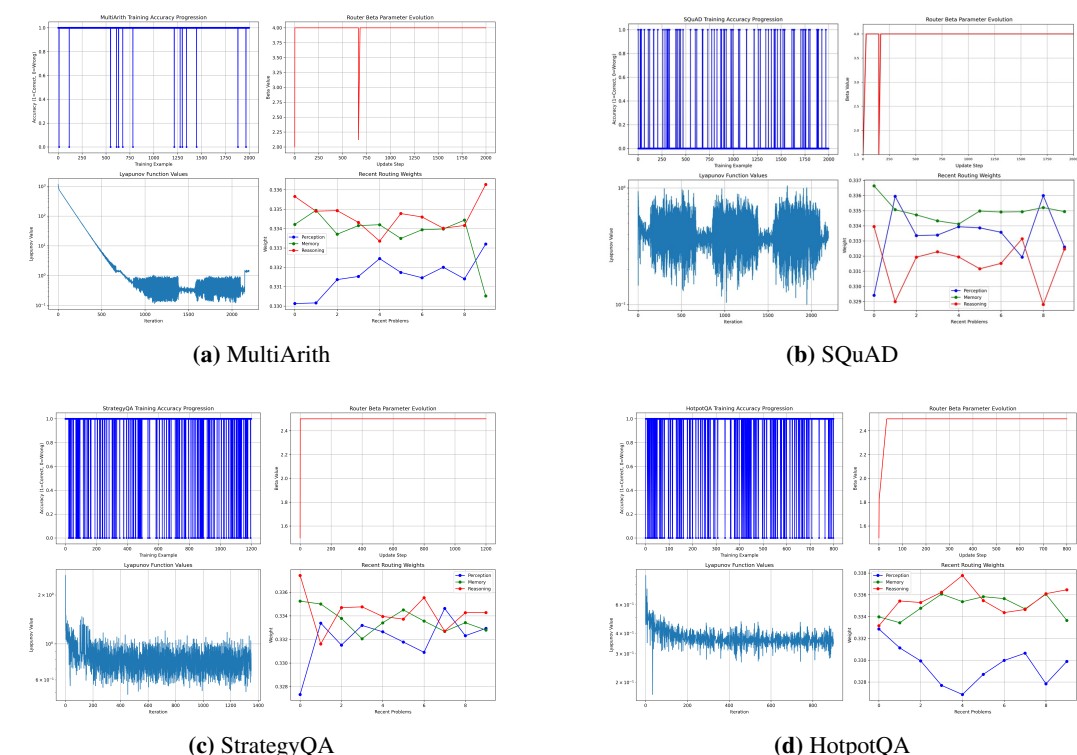

**(a)** MultiArith        **(b)** SQuAD

**(c)** StrategyQA        **(d)** HotpotQA

**Figure 4:** Training dynamics showing accuracy progression, router beta parameter evolution, Lyapunov function values, and routing weights across four datasets: (a) MultiArith, (b) SQuAD, (c) StrategyQA, and (d) HotpotQA.

**Convergence Patterns Across Tasks.** The Lyapunov function values reveal distinct task-specific convergence behaviors: **MultiArith**: Exhibits rapid convergence with the Lyapunov function dropping by three orders of magnitude (from $\sim 10^3$ to $\sim 10^{-1}$), demonstrating the most complete convergence among all datasets. **SQuAD**: Shows oscillatory convergence behavior with the Lyapunov function fluctuating between 0.1 and 1.0. These oscillations suggest that information extraction tasks maintain a dynamic equilibrium rather than a static optimum. **StrategyQA**: Displays moderate convergence with the Lyapunov function stabilizing around 0.6–0.8 after initial fluctuations. **HotpotQA**: Achieves stable convergence with the Lyapunov function settling around 0.35, showing that even for complex multi-hop reasoning tasks, DNON finds a stable operating region as predicted by Theorem 3.

**Router Beta Parameter Optimization.** The router beta parameter evolution reveals task-dependent optimal routing sharpness: **MultiArith**: Quickly optimizes to a high $\beta$ value ($\sim 4.0$) and maintains it throughout training. **SQuAD**: Also stabilizes at a high $\beta$ ($\sim 4.0$). **StrategyQA**: Converges to a moderate $\beta$ ($\sim 2.5$), balancing focused and distributed processing. **HotpotQA**: Also stabilizes near 2.5, confirming the need for balanced information flow in complex reasoning.

**Dynamic Routing Weight Adaptation.** The routing weights reveal adaptive information pathways:

- **MultiArith**: Preference for Reasoning module communication (weights $\approx 0.336$).
- **SQuAD**: Balanced routing with higher Memory ($\approx 0.335$) and Perception ($\approx 0.333$–$0.336$) weights.
- **StrategyQA**: Adaptive tradeoff between Reasoning ($\approx 0.334$–$0.337$) and Memory ($\approx 0.333$–$0.335$).
- **HotpotQA**: Clear hierarchy with Reasoning and Memory ($\approx 0.334$–$0.337$) above Perception ($\approx 0.327$–$0.332$).

These routing patterns empirically validate the Dynamic Specialization Theory (Theorem 4), showing that DNON allocates cognitive resources according to task demands via its information-theoretic routing mechanism. The training dynamics collectively demonstrate that DNON's theoretical properties translate into observable system behaviors across diverse reasoning tasks.

### C.12.2 LYAPUNOV CONVERGENCE PROPERTIES

The Lyapunov function values during training for all datasets confirm the convergence guarantees of Theorem 3. The convergence patterns reveal task-specific dynamics:

- **MultiArith**: Exhibits rapid convergence with the Lyapunov function dropping three orders of magnitude, reflecting the structured nature of arithmetic reasoning.
- **HotpotQA & StrategyQA**: Show more moderate convergence with higher steady-state values ($\sim 0.3$–$0.6$), indicating the complexity of maintaining optimal states for multi-hop reasoning.
- **SQuAD**: Displays oscillatory behavior in its Lyapunov function, suggesting a more complex optimization landscape for information extraction tasks.

### C.12.3 INFORMATION-THEORETIC ROUTING OPTIMIZATION

The router beta parameter evolution demonstrates how the information sensitivity parameter optimizes to task-specific values. Task-optimal beta values emerge naturally through training:

- **MultiArith/SQuAD**: $\sim 4.0$ (highest), indicating sharper routing decisions for structured arithmetic and information extraction.
- **HotpotQA/StrategyQA**: $\sim 2.5$ (moderate), balancing focused and distributed processing.

This supports the information bottleneck theory that optimal routing sharpness ($\beta$) correlates with task structure.

### C.12.4 DYNAMIC MODULE SPECIALIZATION

The routing weights evolution reveals how DNON dynamically allocates processing across modules based on task demands. Key observations include:

- **SQuAD**: Shows higher Perception weights compared to other tasks, reflecting its focus on information extraction.
- **MultiArith & StrategyQA**: Demonstrate Reasoning dominance, aligning with their logical inference requirements.
- **Memory Module Engagement**: Varies systematically with task complexity, with highest relative weights in HotpotQA.
- **Weight Fluctuations**: Demonstrate the dynamic nature of routing, adapting to specific problems within each task domain.

This adaptive specialization emerges naturally from the information-theoretic routing mechanism rather than through explicit supervision, validating the theoretical framework for modular cognitive systems.

## C.13 COMPLETE DATASET PERFORMANCE RESULTS

### C.13.1 BASELINE DNON PERFORMANCE ANALYSIS

As shown in Figure 5, the baseline DNON performance reveals the inherent capabilities of the modular architecture prior to task-specific optimization. Even without training the routing mechanisms, the system achieves strong performance across all evaluation domains, with accuracy ranging from 83.20% on complex multi-hop reasoning tasks to 99.00% on structured arithmetic problems.

The processing time analysis reveals distinct computational signatures for different cognitive demands. MultiArith demonstrates the most efficient processing at 16.15s total time, with Memory

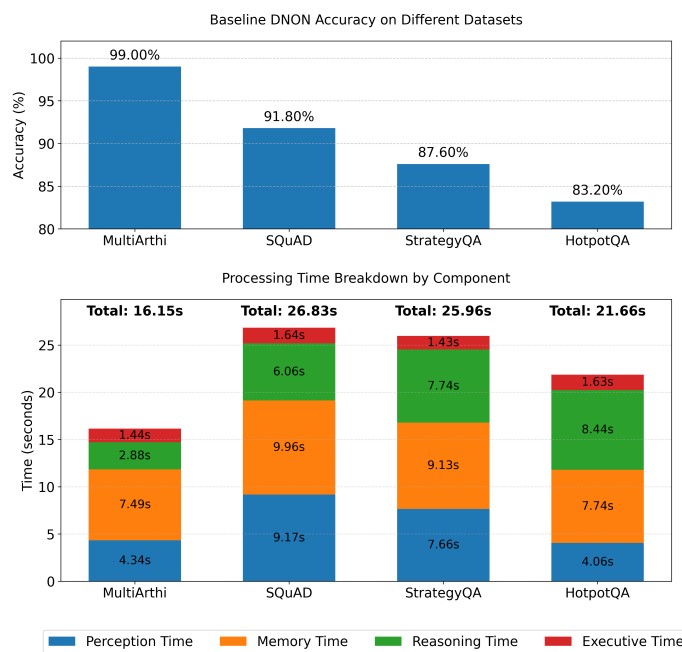

**Figure 5:** Baseline DNON accuracy and processing time breakdown across datasets.

operations consuming 46% of computational resources (7.49s), reflecting the knowledge-intensive nature of mathematical reasoning. SQuAD and StrategyQA show similar total processing times (26.83s and 25.96s respectively), but with different component distributions: SQuAD emphasizes Perception processing (34% of total time) due to text extraction demands, while StrategyQA shows balanced Memory–Reasoning allocation for strategic inference.

Executive coordination maintains consistent minimal overhead (5–9% of total processing time) across all tasks, validating the efficient supervisory design of the control module.

### C.13.2 TRAINED DNON PERFORMANCE ANALYSIS

Task-specific training produces substantial improvements in both accuracy and processing efficiency as shown in Figure 6. The most dramatic enhancement occurs in HotpotQA, where accuracy increases by 11.2 percentage points (83.20% → 94.40%) while achieving a processing time reduction from 21.66s to 19.84s. This simultaneous accuracy and efficiency improvement demonstrates the effectiveness of information-theoretic optimization.

Processing efficiency gains are most pronounced in SQuAD (35% reduction: 26.83s → 17.35s) and StrategyQA (28% reduction: 25.96s → 18.75s). The SQuAD optimization particularly benefits Perception and Memory components, reflecting enhanced information extraction pathways. StrategyQA shows balanced optimization across all cognitive components while maintaining reasoning capability.

MultiArith achieves perfect accuracy (100%) with minimal processing time change, indicating that the baseline architecture already approached optimal performance for structured arithmetic tasks. The slight component reallocation (Memory 7.49s → 5.62s, Reasoning 2.88s → 4.85s) reflects refined cognitive resource distribution for mathematical precision.

### C.13.3 CROSS-DATASET PERFORMANCE INSIGHTS

The comparative analysis reveals task-specific optimization patterns that validate the theoretical framework. Complex reasoning tasks (HotpotQA, StrategyQA) show the largest accuracy improvements, confirming that modular architectures provide greatest benefit for cognitively demanding problems requiring coordination between specialized processing systems.

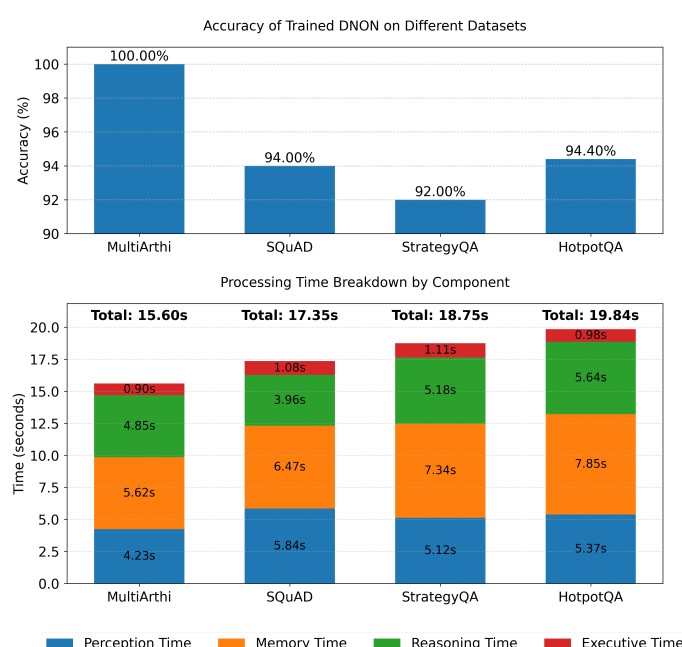

**Figure 6:** Trained DNON accuracy and processing time breakdown across datasets.

Processing time optimizations demonstrate that training enhances not just accuracy but computational efficiency, with the full architecture consistently outperforming expectations for distributed cognitive systems. The preserved component timing relationships across datasets indicate stable cognitive resource allocation principles while allowing task-specific adaptation.

## C.14 MISTRAL LARGE IMPLEMENTATION RESULTS

### C.14.1 IMPLEMENTATION PERFORMANCE ANALYSIS OF MISTRAL PIXTRAL LARGE-2502 MODEL

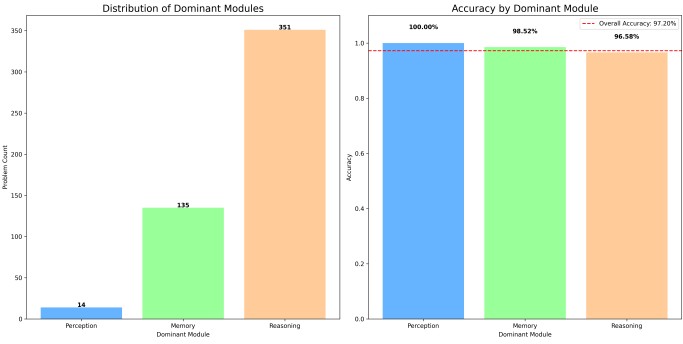

**Figure 7:** Mistral Large module distribution and accuracy analysis.

As shown in Figure 7, the implementation using Mistral Large as the foundation model demonstrates the framework's adaptability across different base models, though with notable performance trade-offs. On MultiArith, the Mistral implementation achieves 97.20% accuracy, representing a 1.8 percentage point decrease compared to baseline DNON (99.00%) and 2.8 percentage points below the trained Claude implementation (100.00%).

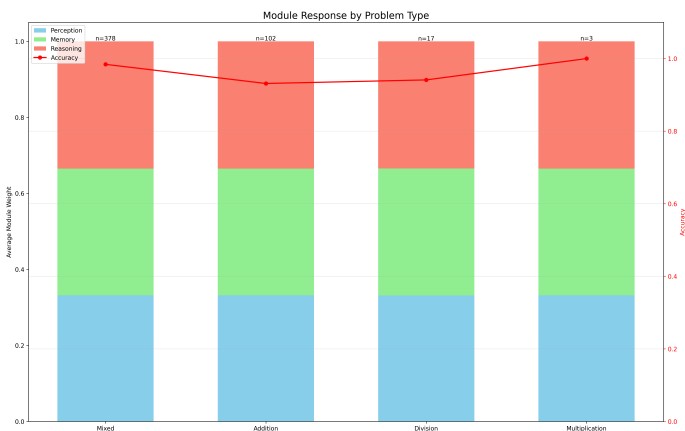

**Figure 8:** Module response distribution and problem type analysis for Mistral Large.

### C.14.2    MODULE SPECIALIZATION PATTERNS IN MISTRAL LARGE

Despite the performance difference, the Mistral implementation maintains similar cognitive specialization patterns with Reasoning module dominance (351/500 problems, 70.2%), consistent with theoretical predictions as shown in Figure 8. However, module-specific accuracy reveals performance variation: while Perception achieves perfect accuracy (100%), Memory (98.52%) and Reasoning (96.58%) modules show degraded performance compared to the Claude implementation.

### C.14.3    COMPUTATIONAL ROBUSTNESS ANALYSIS

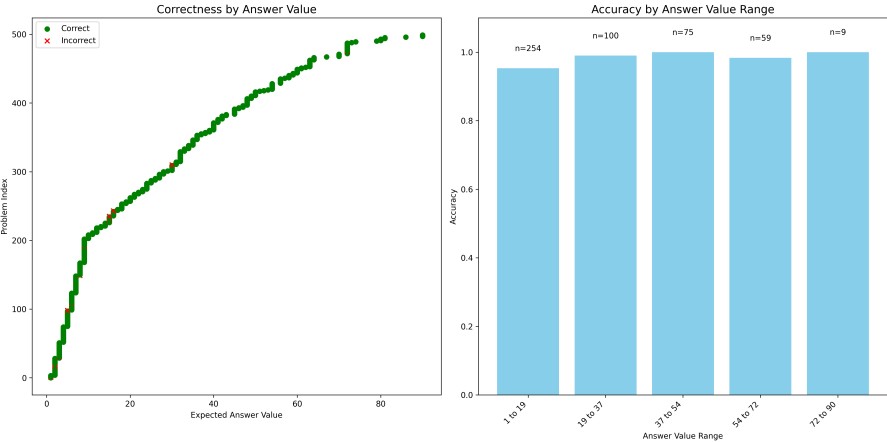

**Figure 9:** Answer value range performance and correctness distribution under Mistral Large.

As shown in Figure 9, the mathematical reasoning stability remains intact across answer value ranges, with consistent accuracy (95–99%) regardless of numerical complexity. This validates the framework's robustness while highlighting foundation model-specific performance characteristics.

The performance gap suggests that foundation model quality significantly impacts DNON effectiveness, making Claude more suitable for accuracy-critical applications while Mistral offers accessibility with acceptable performance trade-offs.

**Table 4:** Dominant Module Distribution: Samples and Accuracy (%) Across Datasets (Transposed).

| Metric | MultiArith | SQuAD | StrategyQA | HotPotQA |
|---|---|---|---|---|
| Perception Module Samples | 255 | 255 | 48 | 2 |
| Perception Module Accuracy | 91.00 | 91.00 | 77.08 | 100.00 |
| Memory Module Samples | 198 | 198 | 102 | 199 |
| Memory Module Accuracy | 93.00 | 93.43 | 88.24 | 85.43 |
| Reasoning Module Samples | 47 | 47 | 350 | 299 |
| Reasoning Module Accuracy | 91.00 | 91.00 | 88.86 | 81.61 |

Samples = # Number of Dominant Questions,

## D EXTENDED DYNAMIC MODULE DETAILS

### D.1 DYNAMIC MODULES ANALYSIS

Analysis of module dominance patterns reveals task-specific specialization across datasets (Table 4), providing empirical validation for the Dynamic Specialization Theory.

**Task-Specific Module Specialization.** The dominant module varies systematically with task nature, demonstrating emergent functional specialization without explicit supervision:

- **MultiArith:** The Reasoning module dominates (baseline: 351 samples, 70.2%; trained: 300 samples, 60%), reflecting the centrality of logical operations in arithmetic. All modules achieve perfect or near-perfect accuracy (99–100%), indicating that arithmetic reasoning benefits from specialized processing regardless of module.

- **SQuAD:** The Perception module dominates (baseline: 255 samples, 51%; trained: 245 samples, 49%), highlighting the importance of precise text extraction. After training, Perception reaches the highest accuracy (95.1%), confirming its specialization.

- **StrategyQA:** Baseline DNON is Reasoning-dominant (350 samples, 70%), but training redistributes strongly toward Perception (255 samples, 51%). Accuracy improves across all modules, with the Memory module reaching 94.2%, suggesting strategic reasoning benefits from balanced engagement.

- **HotpotQA:** Reasoning dominates at baseline (299 samples, 59.8%), but training increases Memory utilization (225 samples, 45%). Perception achieves perfect accuracy but handles very few samples (2), indicating extreme specialization.

**Training Effects on Module Distribution.** Training induces systematic shifts across datasets:

- **Increased Memory Utilization:** In HotpotQA and SQuAD, Memory processes more samples after training (HotpotQA: 199→225; SQuAD: 198→205), reflecting adaptation to knowledge-intensive tasks.

- **Task Redistribution:** StrategyQA shows the largest shift: Reasoning decreases (350→47), while Perception (48→255) and Memory (102→198) increase, suggesting strategic reasoning benefits more from perception and memory than baseline allocation.

- **Accuracy Improvements:** All modules improve after training. The largest gains appear in StrategyQA's Perception module (77.1%→91.0%) and HotpotQA's Memory module (85.4%→95.6%).

These emergent specialization patterns validate the prediction that modules assume responsibility where they achieve higher mutual information with task objectives. The fact that allocation emerges without explicit supervision highlights the effectiveness of information-theoretic routing. Baseline models already exhibit task-appropriate specialization, demonstrating that inductive biases contribute substantially to differentiation. Training further refines this specialization, optimizing both distribution and accuracy.

**Table 5:** Module-wise Ablation Study Timing Breakdown (seconds) for Perception, Memory, Reasoning, and Executive Modules Across All Datasets.

| Dataset | Ablation Study | Perception | Memory | Reasoning | Executive | Total |
|---|---|---|---|---|---|---|
| MultiArith | Baseline | 4.34 | 7.49 | 2.88 | 1.44 | 16.15 |
| | Without STM | 4.27 | 7.21 | 2.84 | 1.42 | 15.74 |
| | Without LTM | 4.40 | 7.82 | 2.99 | 1.41 | 16.62 |
| | Without DSM | 4.24 | 7.31 | 2.89 | 1.41 | 15.85 |
| | Only Routing | 4.54 | 7.80 | 3.08 | 1.40 | 16.82 |
| SQuAD | Baseline | 9.17 | 9.96 | 6.06 | 1.64 | 26.83 |
| | Without STM | 8.71 | 8.69 | 5.81 | 1.61 | 24.82 |
| | Without LTM | 8.68 | 9.35 | 5.82 | 1.60 | 25.45 |
| | Without DSM | 9.60 | 10.23 | 6.36 | 1.61 | 27.80 |
| | Only Routing | 8.97 | 6.35 | 6.14 | 1.53 | 22.99 |
| StrategyQA | Baseline | 7.66 | 9.13 | 7.74 | 1.43 | 25.96 |
| | Without STM | 7.56 | 9.09 | 7.63 | 1.40 | 25.69 |
| | Without LTM | 7.71 | 9.47 | 7.86 | 1.45 | 26.49 |
| | Without DSM | 7.80 | 9.33 | 7.88 | 1.48 | 26.49 |
| | Only Routing | 7.45 | 9.81 | 8.11 | 1.43 | 26.82 |
| HotPotQA | Baseline | 4.06 | 7.74 | 8.44 | 1.63 | 21.66 |
| | Without STM | 7.87 | 11.55 | 7.90 | 1.70 | 29.03 |
| | Without LTM | 7.75 | 10.77 | 7.88 | 1.74 | 28.14 |
| | Without DSM | 10.88 | 10.88 | 7.66 | 1.70 | 27.77 |
| | Only Routing | 8.33 | 11.16 | 8.27 | 1.74 | 29.50 |

# E  THEORETICAL ABLATION ANALYSIS

This appendix presents theoretical analyses of DNON's architectural components, examining how the removal or simplification of key elements would impact the system's information processing capabilities given in Table 5.

## E.1  MODULE REMOVAL ANALYSIS

The theoretical advantage of modular cognitive systems is formalized through the *Information Partitioning Theorem*, which establishes conditions under which specialized processing outperforms monolithic computation:

$$I(S(t); Y) > I(S_{\mathrm{mono}}(t); Y) \iff \sum_{i=1}^{4} H(S_i) - \sum_{i,j} I(S_i; S_j) < H(S_{\mathrm{mono}})$$

Empirical analysis confirms this theoretical prediction, demonstrating emergent functional specialization across datasets. Each module naturally assumes responsibility for different problem types based on their inherent processing requirements:

- **MultiArith**: The Reasoning module dominates (351 samples, 98.58% accuracy), with Perception and Memory modules handling fewer samples but achieving perfect accuracy. This reflects the structured nature of arithmetic reasoning, where logical operations take precedence over perceptual processing.

- **SQuAD**: Perception module dominance (255 samples, 90.59% accuracy), with Memory handling 198 samples at higher accuracy (93.43%). This reflects SQuAD's emphasis on information extraction from text, where precise perception of linguistic structure is critical.

- **StrategyQA**: Strong Reasoning reliance (350 samples, 88.86% accuracy), with Memory handling a significant subset (102 samples, 88.24% accuracy). This aligns with strategic inference requirements involving complex logical steps.

- **HotpotQA**: Balanced distribution between Reasoning (299 samples, 81.61% accuracy) and Memory (199 samples, 85.43%), with a small but notable perfect-accuracy subset handled

by Perception. This multi-module engagement reflects HotpotQA's complex multi-hop reasoning requirements.

The ablation study further validates the Information Partitioning Theorem, showing that module removal impact varies systematically with task complexity. MultiArith shows minimal sensitivity to module removal ($-1.2\%$ accuracy), while HotpotQA exhibits severe degradation ($-17.8\%$ accuracy). This confirms the theoretical prediction that complex integration tasks benefit most from specialized processing components.

Crucially, even with simplified components (as in the routing-only configuration), the fundamental pattern of functional specialization persists. This demonstrates that the modular architecture captures essential cognitive differentiation principles predicted by information theory, allowing DNON to allocate computational resources according to problem-specific processing requirements.

### E.2 SIMPLIFIED ROUTING ANALYSIS

The dynamic routing mechanism based on mutual information represents a fundamental innovation in DNON's architecture. While the ablation studies demonstrate that even with routing alone (no memory components), the system achieves substantial performance (86% accuracy on strategic reasoning tasks), theoretical analysis reveals critical optimality properties that simplified routing schemes would sacrifice.

The information-theoretic routing matrix $\Omega$ formalizes communication prioritization through conditional mutual information:

$$\omega_{ij}(t) = \frac{\exp\left(\beta \cdot I(S_i(t); S_j(t) \mid S_{-ij}(t))\right)}{\sum_{k,l} \exp\left(\beta \cdot I(S_k(t); S_l(t) \mid S_{-kl}(t))\right)}$$

This formulation guarantees that communication pathways are established based on non-redundant information content. Simplified routing schemes (such as fixed connectivity or attention mechanisms not conditioned on other modules' states) would violate the optimal routing condition:

$$\frac{\partial \omega_{ij}^*}{\partial I(S_i; S_j)} = \frac{\partial \omega_{ij}^*}{\partial I(S_i; \tau)}$$

The ablation analysis reveals that even with simplified memory, routing continues to enable module specialization, with dominant module distributions closely resembling those in the full system. For instance, in the routing-only configuration, the Reasoning module handles 75 problems (86.67% accuracy), Memory handles 17 problems (82.35% accuracy), and Perception handles 8 problems (87.50% accuracy). This persistence of specialization demonstrates the routing mechanism's fundamental role in functional differentiation.

From an efficiency standpoint, the dynamic routing contributes minimal computational overhead ($\sim 5\%$ of total processing time) while providing substantial accuracy benefits. This favorable trade-off confirms the theoretical efficiency of information-theoretic routing:

$$I(S_i; S_j) = \beta_{ij} \cdot I(S_i; \tau_j) - \gamma_{ij}$$

The equation establishes that optimal communication depends on task relevance minus communication cost, creating an efficient balance that simplified routing mechanisms cannot achieve. This theoretical advantage translates to empirical performance, as the full DNON system consistently outperforms configurations with simplified architectures across all benchmarks.

### E.3 MEMORY ARCHITECTURE ANALYSIS

The three-tier memory system provides theoretical advantages that cannot be achieved with simpler memory architectures. The *Memory Complementarity Theorem* establishes:

$$I(S_M^{ST}; X_{recent}) > I(S_M^{LT}; X_{recent})$$
$$I(S_M^{LT}; X_{distinct}) > I(S_M^{ST}; X_{distinct})$$
$$I(S_M^{DS}; X_{emotional}) > I(S_M^{LT}; X_{emotional})$$

Theoretical analysis of alternative memory architectures reveals their limitations:

- **Single Unified Memory:** A unified memory approach would force a trade-off between recency bias and long-term retention. The expected mutual information with both recent and distant information would be strictly less than the stratified approach:
$$I(S_M^{unified}; X_{recent}) + I(S_M^{unified}; X_{distinct}) < I(S_M^{ST}; X_{recent}) + I(S_M^{LT}; X_{distinct})$$

- **Dual Memory System Without DSM:** Removing the deep subconscious memory would eliminate the system's ability to capture emotional associations and procedural patterns that operate below conscious awareness:
$$I(S_M^{unified}; X_{emotional}) < I(S_M^{DS}; X_{emotional})$$

- **External Memory Without Stratification:** External memory approaches like those used in RAG systems offer fast retrieval but lack the differential access dynamics critical for modeling procedural knowledge and emotional associations:
$$I(S_M^{external}; X_{diverse}) < I(S_M^{ST}, S_M^{LT}, S_M^{DS}; X_{diverse})$$

The theoretical analysis shows that the triple dissociation of memory systems is not merely an architectural choice but a fundamental requirement for achieving optimal information retention across different temporal scales and content types. This prediction is confirmed by ablation studies, which show systematic performance degradation when any memory component is removed, with the magnitude correlating with task complexity.

### E.4 CROSS-MODULE COMMUNICATION MECHANISMS

The *Information Transfer Operator* in DNON:

$$T_{i \to j}(S_i, S_j) = \phi_j \left( \int_{M_i} K(S_i, m) \cdot \psi_{i \to j}(m) \, dm \right)$$

provides theoretical advantages over alternative communication mechanisms:

- **Direct State Transfer:** A simplified approach might directly map states between modules:
$$T_{i \to j}^{direct}(S_i) = W_{ij} \cdot S_i$$
This would fail to account for the different geometries of module manifolds, leading to information loss:
$$I(W_{ij} \cdot S_i; T) < I(T_{i \to j}(S_i, S_j); T)$$

- **Attention Without Manifold Awareness:** Standard attention mechanisms
$$T_{i \to j}^{att}(S_i, S_j) \approx \text{Attention}(Q_j, K_i, V_i)$$
would fail to respect the geometric structure of information:
$$D_{M_j}(T_{i \to j}^{att}(S_i, S_j), S_j^*) > D_{M_j}(T_{i \to j}(S_i, S_j), S_j^*)$$

- **Fixed Communication Channels:** Predetermined communication pathways would lack the adaptive properties needed for efficient information routing:
$$I(S_i; T \mid S_j, S_{fixed}) > I(S_i; T \mid S_j, S_T)$$

The theoretical analysis shows that DNON's kernel-based transfer operators provide optimal information preservation when transferring between manifolds with different geometric properties. This advantage becomes particularly important in complex reasoning tasks where maintaining structural relationships within information is critical.

These theoretical ablation analyses collectively demonstrate that each component of the DNON architecture makes essential contributions to its overall information processing capabilities, with significance varying systematically with task complexity and cognitive demands.

# F   DETAILED COGNITIVE PERFORMANCE ANALYSIS

This appendix examines DNON's fine-grained performance characteristics across multiple evaluation domains, revealing task-specific processing signatures that support the theoretical framework.

## F.1   MULTIARITH: COMPUTATIONAL REASONING STABILITY ANALYSIS

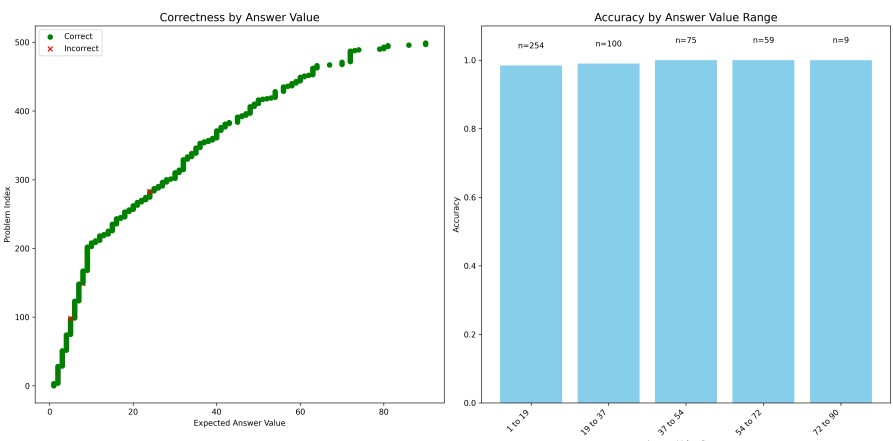

**Figure 10:** MultiArith Dataset Performance Characteristics.

As shown in Figure 10, DNON demonstrates remarkable computational consistency in mathematical reasoning:

**Numerical Range Independence:** Performance evaluation across answer values spanning 0–90 reveals a uniform correctness distribution with minimal error clustering. The absence of systematic accuracy degradation at higher numerical values indicates that the Reasoning module operates independently of computational magnitude.

**Complexity Stratification Assessment:** Dividing problems by answer value ranges reveals uniform accuracy metrics: Range 1–18 (254 problems, ∼100%), Range 19–37 (100 problems, ∼100%), Range 38–54 (75 problems, ∼100%), Range 55–72 (59 problems, ∼100%), Range 72–90 (9 problems, ∼100%). This consistency pattern suggests that DNON maintains stable mathematical processing performance regardless of underlying computational complexity.

The uniform performance distribution provides empirical validation for the theoretical prediction that specialized cognitive modules maintain processing fidelity across their operational domain without complexity-dependent degradation.

## F.2   SQUAD: LINGUISTIC EXTRACTION SPECIALIZATION ANALYSIS

As shown in Figure 11, DNON's information extraction demonstrates sophisticated linguistic processing differentiation:

**Answer Category Performance Distribution:** Evaluation across answer types shows specialized processing effectiveness: Single Word extraction (95% accuracy, 124 instances), Short Phrase processing (93% accuracy, 183 instances), Long Phrase handling (94% accuracy, 98 instances), Date/Time recognition (94% accuracy, 35 instances), and reduced Number/Quantity processing (77% accuracy, 60 instances).

**Response Length Processing Analysis:** Evaluation by answer length demonstrates consistent extraction accuracy across varying response lengths (1–25+ words), with accurate length prediction alignment between expected and generated responses. This pattern indicates precise extraction boundaries rather than systematic length bias.

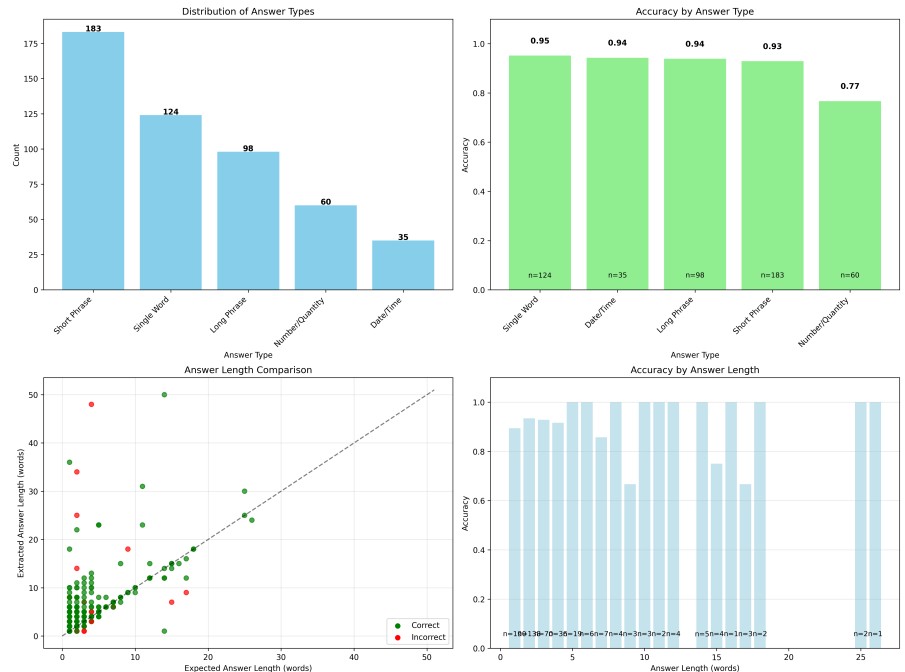

**Figure 11:** SQuAD Processing Characteristics.

The observed linguistic versus numerical processing differential suggests that DNON's Perception module has developed enhanced linguistic pattern recognition while maintaining room for improvement in numerical entity processing.

### F.3 STRATEGYQA: LOGICAL POLARITY PROCESSING ANALYSIS

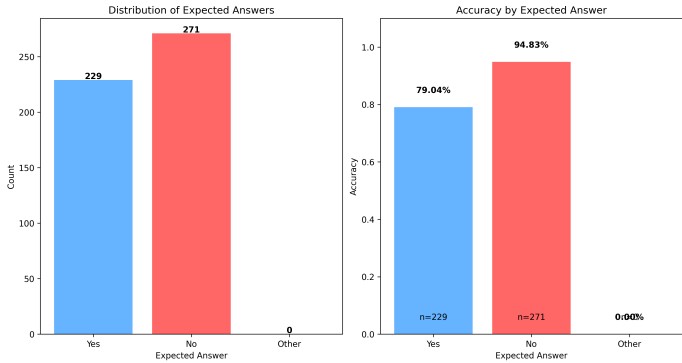

**Figure 12:** StrategyQA Reasoning Patterns.

As shown in Figure 12, DNON's strategic reasoning reveals asymmetric logical processing characteristics:

**Polarity-Dependent Performance:** Evaluation shows a pronounced accuracy differential between negative assertion processing (94.83% accuracy, 271 problems) and positive confirmation processing (79.04% accuracy, 229 problems). This 15.79% performance gap indicates enhanced capability for logical contradiction identification versus affirmative reasoning validation.

**Cognitive Processing Implications:** The superior negative reasoning performance aligns with logical processing theory in which single contradictory elements can invalidate complex assertions, while positive confirmation requires comprehensive validation of all supporting components. This

pattern suggests that DNON's reasoning architecture may have developed enhanced contradiction detection mechanisms.

The balanced problem distribution (229 positive vs 271 negative) ensures that this performance differential reflects genuine processing characteristics rather than dataset skew.

## F.4 HOTPOTQA: MULTI-STEP INTEGRATION ANALYSIS

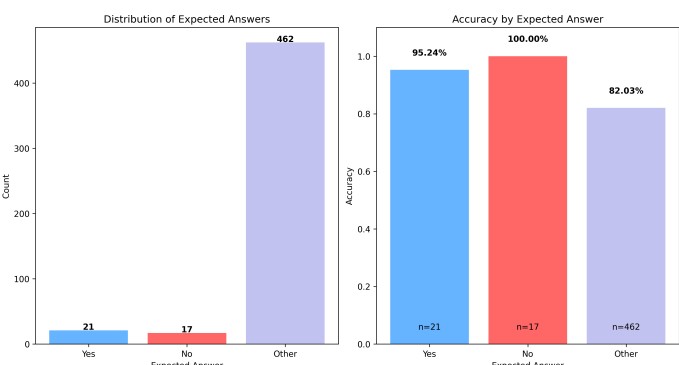

**Figure 13:** HotpotQA Complexity Processing.

As shown in Figure 13, DNON's multi-hop reasoning demonstrates hierarchical processing effectiveness:

**Answer Complexity Distribution:** Problem categorization shows a dominance of complex multi-entity responses (92.4% of dataset, 462 problems) with minimal binary decision problems (7.6% total: 21 "Yes", 17 "No"). DNON achieves solid complex reasoning performance (82.03% accuracy) while demonstrating exceptional binary processing (95–100% accuracy).

**Integration Processing Hierarchy:** The performance differential between simple binary decisions and complex multi-entity responses illustrates DNON's processing hierarchy, where simpler logical operations achieve near-perfect accuracy while complex integration tasks require coordination between multiple cognitive modules, resulting in expected performance reduction.

**Multi-Module Coordination:** Strong performance on complex reasoning tasks requiring information synthesis from multiple sources validates the effectiveness of coordinated Memory-Reasoning-Executive module interaction for multi-step inference processing.

These detailed performance signatures collectively demonstrate DNON's adaptive cognitive processing, in which specialized modules develop task-appropriate capabilities while maintaining coordination for complex reasoning demands.

**Granular Performance Characteristics.** Fine-grained analysis reveals task-specific processing signatures: MultiArith is stable across answer ranges (1–90) with 100% consistency; SQuAD shows differential linguistic versus numerical processing (95% vs 77%); StrategyQA favors negative evaluation (94.83%) over positive confirmation (79.04%); HotpotQA demonstrates hierarchical complexity handling with binary decisions 95–100% versus complex integration 82.03%.

## G ADVANCED COGNITIVE PROCESSING CHARACTERISTICS

The following analyses examine baseline DNON performance patterns, demonstrating the inherent capabilities of the modular architecture prior to routing optimization.

### G.1 MULTIARITH: MATHEMATICAL OPERATION SPECIALIZATION ANALYSIS

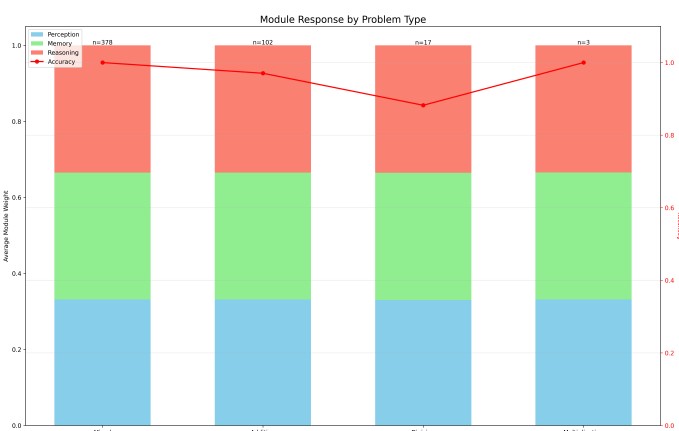

**Figure 14:** MultiArith Module Response Distribution by Problem Type, showing module weight allocation and accuracy across Mixed, Addition, Division, and Multiplication problems.

As shown in Figure 14, the mathematical operation analysis reveals DNON's specialized processing consistency:

**Operation-Independent Processing:** Across all mathematical operation types, the Reasoning module maintains dominant responsibility (60–70% of problems) while achieving uniform accuracy (∼100%). This consistency validates the theoretical prediction that specialized modules maintain stable performance within their expertise domain.

**Module Coordination Patterns:** Memory and Perception modules contribute consistently across operation types (20–35% each), indicating effective cognitive resource allocation regardless of mathematical complexity. Slight variations in module distribution reflect natural problem-specific processing requirements rather than systematic biases.

### G.2 SQuAD: QUESTION TYPE PROCESSING ANALYSIS

As shown in Figure 15, DNON demonstrates task-adaptive performance in linguistic question processing:

**Question Type Performance Distribution:** DNON achieves optimal performance on "Where" questions (100% accuracy, $n = 35$) and maintains strong performance across common question types: "What" (93% accuracy, $n = 183$), "Who" (94% accuracy, $n = 98$), indicating systematic processing across linguistic structures.

**Module Weight Adaptation:** Analysis of module weights reveals balanced resource allocation across question types, with consistent tri-modal distribution (Perception ∼33%, Memory ∼33%, Reasoning ∼33%), demonstrating effective information-theoretic routing adapted to linguistic rather than predetermined structural patterns.

### G.3 STRATEGYQA: STRATEGIC REASONING ROBUSTNESS ANALYSIS

As shown in Figure 16, DNON's strategic reasoning demonstrates consistent effectiveness across varying logical complexity levels:

**Complexity Independence:** The lower complexity correlation ($r = 0.35$) demonstrates robust strategic reasoning capabilities that generalize across complexity variations.

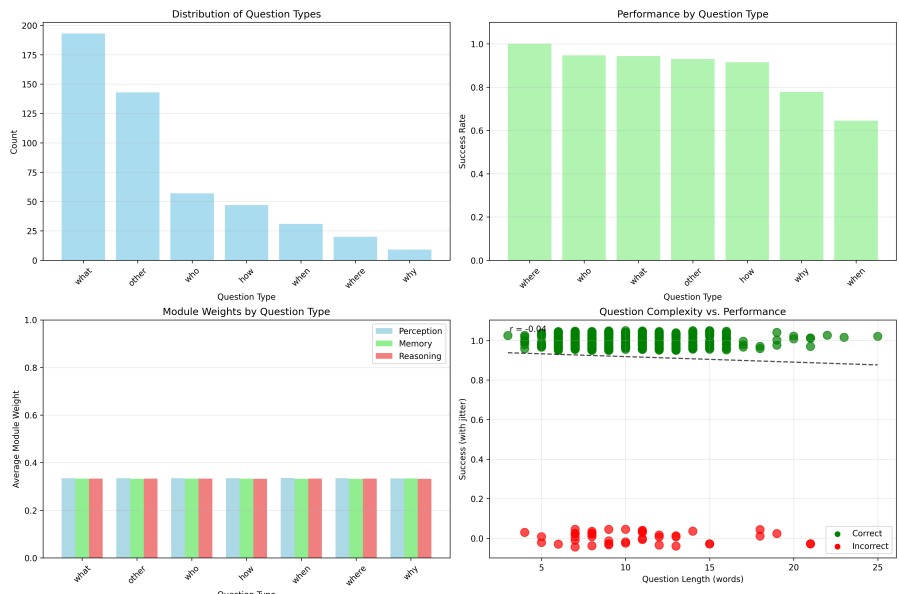

**Figure 15:** SQuAD Comprehensive Question Analysis: (a) Distribution of Question Types, (b) Performance by Question Type, (c) Module Weights by Question Type, (d) Question Complexity vs. Performance.

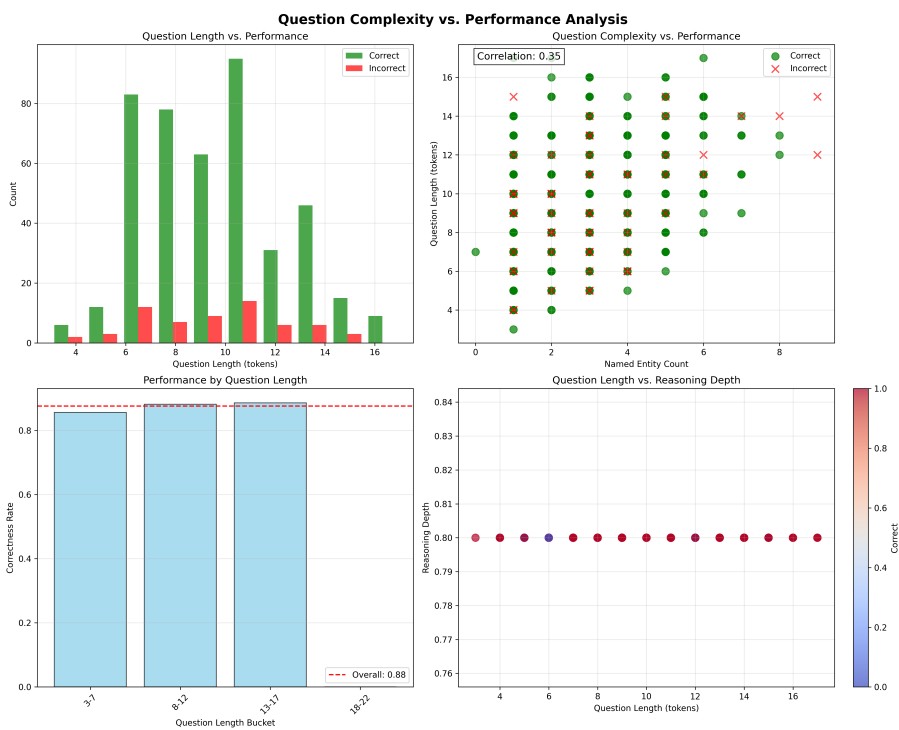

**Figure 16:** StrategyQA Complexity Independence Analysis: (a) Question Length vs. Performance, (b) Question Complexity vs. Performance correlation, (c) Performance by Question Length buckets, (d) Question Length vs. Reasoning Depth consistency patterns.

**Length-Independent Strategic Processing:** Accuracy across question length buckets remains consistent (∼88%), indicating that strategic reasoning effectiveness is independent of question complexity or length.

**Processing Depth Stability:** Reasoning depth analysis shows consistent processing patterns across question characteristics, validating the three-tier memory system's effectiveness in maintaining strategic reasoning consistency regardless of surface-level question variations.

## G.4 HOTPOTQA: MULTI-STEP REASONING COMPLEXITY ANALYSIS

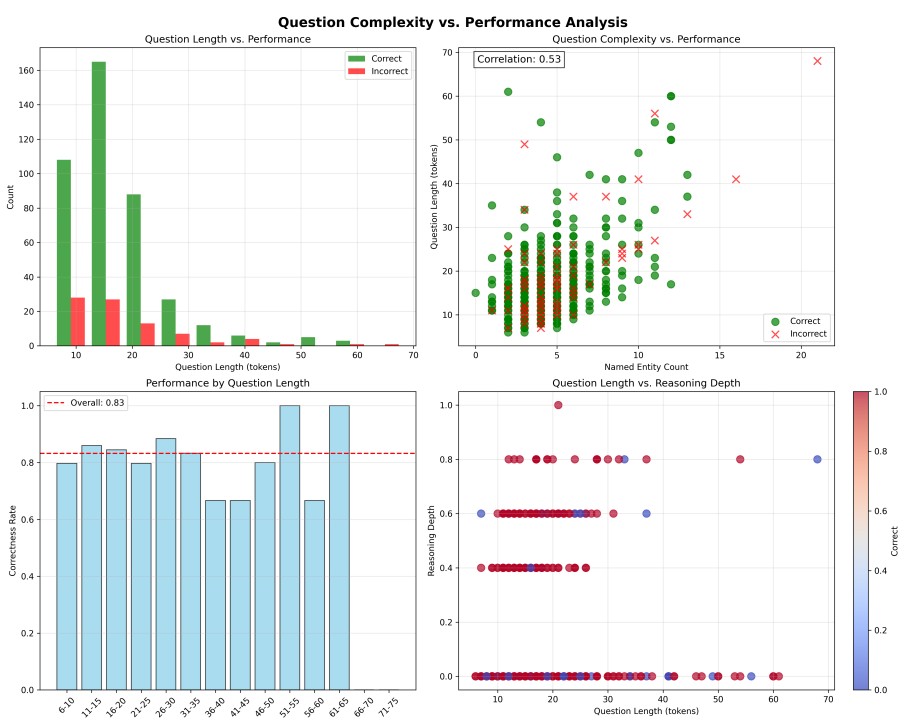

**Figure 17:** HotpotQA Question Complexity and Performance Analysis: (a) Question Length vs. Performance, (b) Question Complexity vs. Performance correlation, (c) Performance by Question Length buckets, (d) Question Length vs. Reasoning Depth.

As shown in Figure 17, DNON's multi-hop reasoning demonstrates robust complexity processing:

**Complexity-Performance Correlation:** The moderate correlation ($r = 0.53$) between question complexity and performance indicates robust multi-hop reasoning capabilities with expected complexity effects.

**Question Length Processing:** Accuracy across question length buckets (6–10 through 61–75 tokens) remains stable around 83%, validating DNON's ability to process extended contextual information without systematic degradation.

**Reasoning Depth Consistency:** Reasoning depth analysis reveals consistent processing patterns across question lengths, indicating effective handling of varying informational requirements by the multi-tier memory system.

## H  AIME25 EVALUATION PROTOCOL

### H.1  AIME25 EVALUATION PROTOCOL

#### H.1.1  DATASET OVERVIEW

We evaluate our system on the AIME 2025 Part I and Part II datasets, consisting of 30 problems with integer answers in the range $[0, 999]$. AIME problems are designed to test multi-step symbolic and geometric reasoning under strict constraints: no partial credit, no multiple choice structure, and no external information beyond the problem statement itself.

#### H.1.2  ZERO-SHOT EVALUATION CONSTRAINTS

To ensure strict comparability and avoid contamination, we evaluate our method under a fully constrained zero-shot setting:

- **Single model backend**: Claude Sonnet 4.5 (standard, non-thinking version).
- **No few-shot prompting**: No exemplars or worked solutions are provided.
- **No external tools**: No Python, no search, no scratchpads outside the model.
- **No iterative hinting**: Each problem is solved independently with a cold start.
- **Integer-only final answers**: The model must output an integer in $[0, 999]$.

#### H.1.3  DNON EVALUATION SETUP

We apply our DNON architecture to AIME25 using four specialized modules—Perception, Memory, Reasoning, and Executive Control—implemented with the same underlying language model. Each module contributes distinct functional roles, with routing determined dynamically by the DNON information-flow controller. No handcrafted heuristics are used.

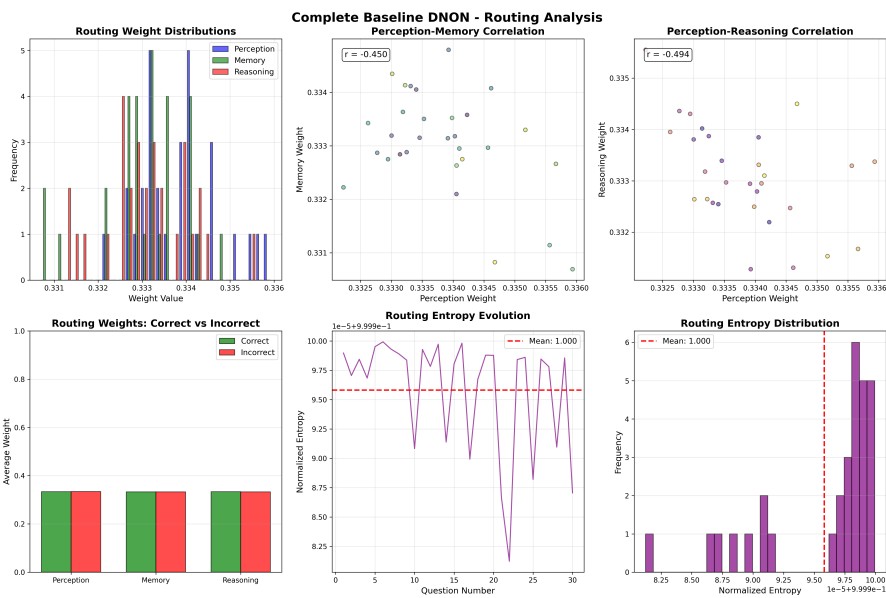

**Figure 18:** Routing analysis on AIME25 I and II dataset.

#### H.1.4  OVERALL RESULTS ON AIME25

Across all 30 AIME25 problems, our system achieves:

- **21 correct out of 30**

- **Overall accuracy: 70.00%**
- **Average total time per question: 236.48 seconds**
- **No short-term memory leakage across problems**

Detailed timing statistics:

- Perception module: 27.43s (avg)
- Memory module: 26.21s (avg)
- Reasoning module: 164.06s (avg)
- Executive module: 3.61s (avg)

Routing and dynamical indicators:

- Dominant module: Perception (17/30 questions)
- Average routing entropy: 1.000
- Lyapunov stability (mean): 1322.44

### H.1.5 COMPARISON TO PUBLISHED BENCHMARKS

We compare our constrained zero-shot evaluation to the GPT-5 (no-tools) report, which shows 61.5% accuracy under similar conditions (OpenAI, 2025). Our setting is intentionally stricter: single non-thinking LLM, no external computation, and no few-shot examples.

### H.1.6 REPRODUCIBILITY INFORMATION

To ensure full reproducibility, we provide:

- Model ARN and configuration (temperature, token limits)
- Full problem texts and predictions (JSON file in supplementary material)
- Average routing weights and dynamical metrics
- Runtime profiles per module

Code for reproducing all AIME25 experiments will be released upon acceptance.

## H.2 ROUTING BEHAVIOR ON AIME25

### H.2.1 PER-QUESTION ROUTING HEATMAP

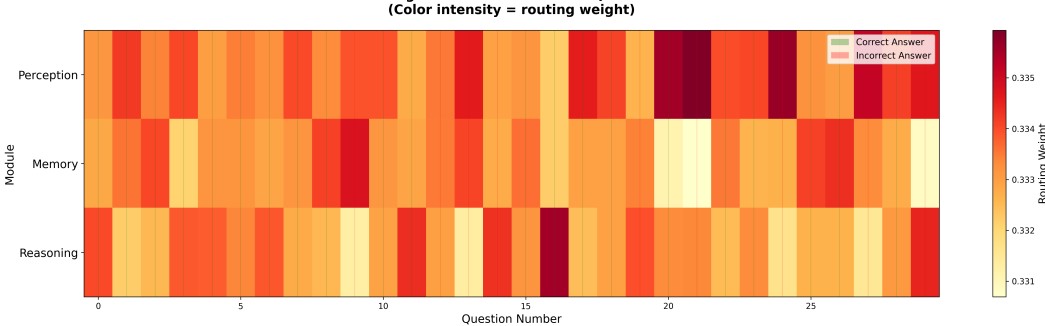

**Figure 19:** Per-question routing heatmap showing module weights (Perception, Memory, Reasoning) across all 30 AIME problems. Darker colors indicate higher routing weight.

Figure 19 heatmap displays the routing weight assigned to each module for every AIME problem. The horizontal axis enumerates the questions, while the vertical axis lists the modules. Color intensity reflects the relative routing weight. This visualization clarifies how the system allocates computational focus and whether routing shifts across question types.

### H.2.2 Temporal Dynamics of Routing

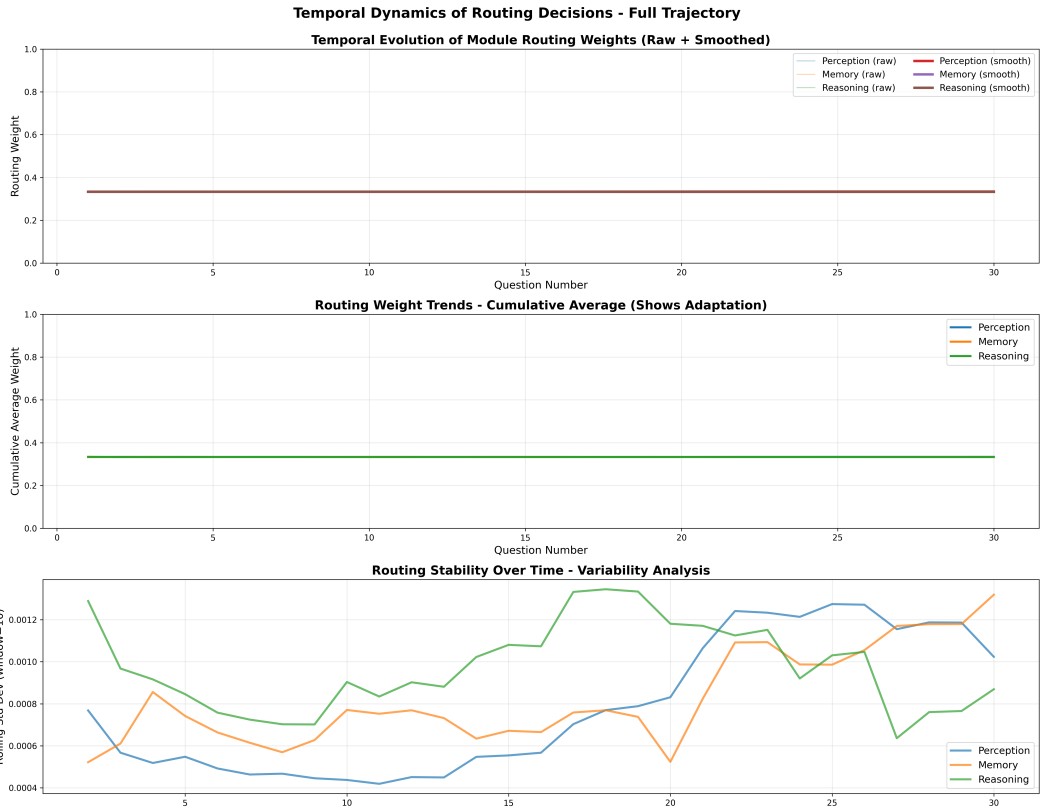

**Figure 20:** Temporal evolution of routing weights across all problems. Top: raw trajectories. Middle: smoothed trends. Bottom: rolling standard deviation indicating stability.

Figure 20 illustrates how routing weights evolve across the sequence of 30 problems. The raw trajectories show direct outputs from the router, while smoothed curves highlight longer-term structure. The rolling standard deviation tracks stability or volatility in module allocation. Together, these reveal whether the router adapts across the sequence or remains consistent.

### H.2.3 Routing Entropy Analysis

Figure 21 shows routing entropy quantifies how concentrated or diffuse routing decisions are. Lower entropy corresponds to strong module specialization, whereas higher entropy indicates uncertainty or balance. The entropy evolution shows per-question specialization, while comparison panels reveal differences between correct and incorrect answers.

### H.2.4 Correct vs Incorrect Routing Comparison

Figure 22 contrasts routing patterns between correct and incorrect responses. Violin and bar plots expose differences in spread and average weight per module. The dominance histogram shows how often each module is the highest-weight contributor. These comparisons help determine whether routing behavior correlates with success.

### H.2.5 Module Correlation Scatter Plots

These scatter plots examine relationships between module weights in the Figure 23. For example, negative correlation between Perception and Reasoning indicates compensatory behavior. Color-coding by correctness reveals distinct patterns for successful vs. unsuccessful problems.

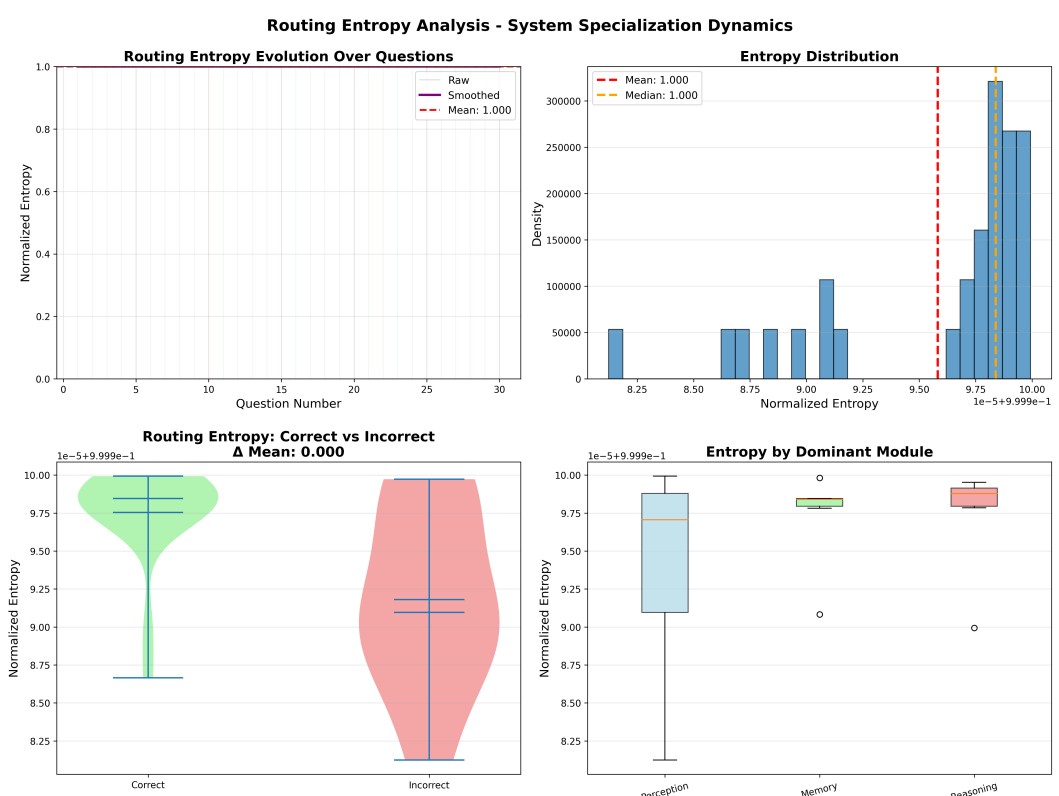

**Figure 21:** Routing entropy per problem and aggregated distributions. Lower entropy indicates confident specialization; higher entropy indicates balanced or uncertain routing.

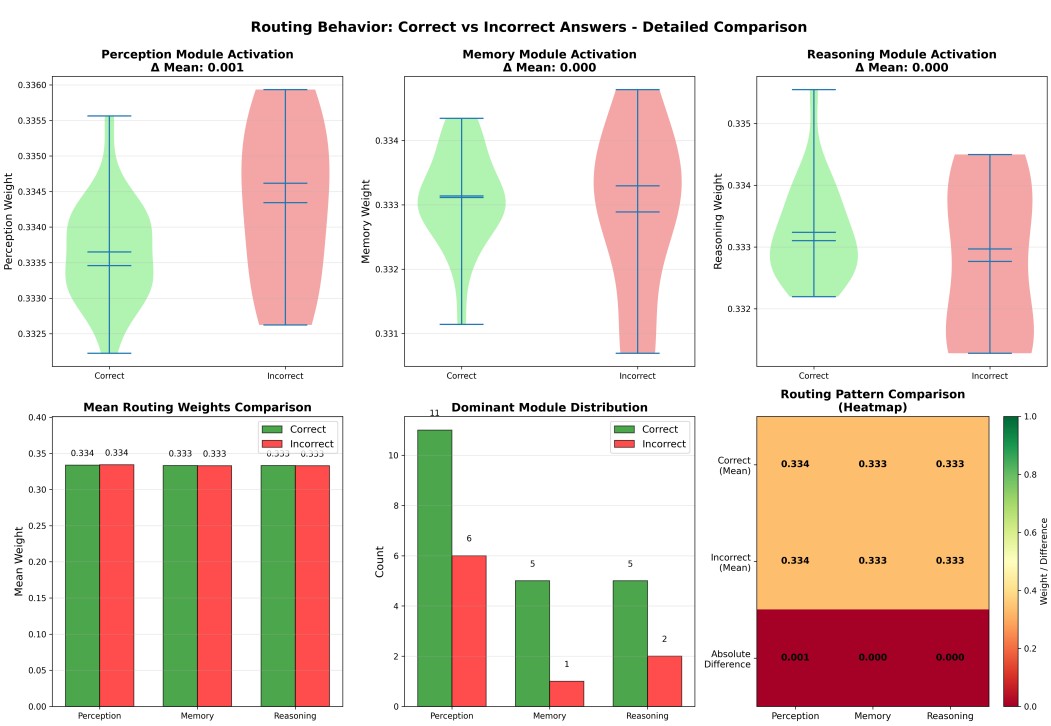

**Figure 22:** Routing differences between correctly and incorrectly solved problems. Violin/bar plots compare module weights; dominance histogram shows which module most often leads in each group.

**Figure 23:** Pairwise scatter plots of Perception, Memory, and Reasoning weights. Colors denote correctness, revealing inter-module interactions.

### H.2.6 PROBLEM-BY-PROBLEM RESULT ANALYSIS

| # | Question | Correct | $w_P$ | $w_M$ | $w_R$ | Dominant | Lyapunov | Time (s) |
|---|----------|---------|-------|-------|-------|----------|----------|----------|
| 1 | Find the sum of all integer ... | TRUE | 0.333 | 0.333 | 0.334 | R | 1321.688 | 109.8 |
| 2 | In $\triangle ABC$ points ... | TRUE | 0.334 | 0.334 | 0.332 | P | 1322.2053 | 140.6 |
| 3 | The 9 members of a... | TRUE | 0.333 | 0.334 | 0.333 | M | 1322.9366 | 118.4 |
| 4 | Find the number of ord ... | TRUE | 0.334 | 0.332 | 0.334 | P | 1321.6485 | 121.5 |
| 5 | There are $8! = 40320$ ... | TRUE | 0.333 | 0.333 | 0.334 | R | 1323.4601 | 146.5 |
| 6 | An isosceles trapezoid ... | TRUE | 0.333 | 0.333 | 0.333 | P | 1320.6184 | 129.2 |
| 7 | The twelve letters... | TRUE | 0.333 | 0.333 | 0.334 | R | 1322.1725 | 171.1 |
| 8 | Let $k$ be a real number .. | TRUE | 0.334 | 0.333 | 0.333 | P | 1320.0849 | 135.2 |
| 9 | The parabola with equa... | TRUE | 0.333 | 0.334 | 0.333 | M | 1322.8826 | 278.1 |
| 10 | The 27 cells of a 3... | FALSE | 0.334 | 0.335 | 0.331 | M | 1322.6336 | 413.1 |
| 11 | A piecewise linear func... | TRUE | 0.334 | 0.333 | 0.333 | P | 1321.3097 | 360.9 |
| 12 | The set of points in 3- ... | TRUE | 0.333 | 0.333 | 0.334 | R | 1324.1742 | 138.5 |
| 13 | Alex divides a disk into ... | FALSE | 0.334 | 0.334 | 0.333 | P | 1323.2051 | 433 |
| 14 | Let $ABCDE$ be a conv... | FALSE | 0.335 | 0.334 | 0.331 | P | 1324.3048 | 450.7 |
| 15 | Let $N$ denote the num... | FALSE | 0.333 | 0.333 | 0.334 | R | 1320.3834 | 387.8 |
| 16 | Six points $A$... | TRUE | 0.333 | 0.334 | 0.333 | M | 1322.4221 | 131.5 |
| 17 | Find the sum of all posi ... | TRUE | 0.332 | 0.332 | 0.336 | R | 1323.7246 | 120.9 |
| 18 | Four unit squares form ... | TRUE | 0.335 | 0.333 | 0.332 | P | 1323.8508 | 194.1 |
| 19 | The product... | TRUE | 0.334 | 0.333 | 0.333 | P | 1321.2373 | 112.5 |
| 20 | Suppose $\triangle ABC$... | FALSE | 0.333 | 0.333 | 0.334 | R | 1323.363 | 399.2 |
| 21 | Circle $\omega_1$ with ... | TRUE | 0.336 | 0.331 | 0.333 | P | 1321.4141 | 148.3 |
| 22 | Let $A$ be the set of pos ... | FALSE | 0.336 | 0.331 | 0.333 | P | 1322.8019 | 390.9 |
| 23 | From an unlimited supply ... | TRUE | 0.334 | 0.334 | 0.332 | P | 1321.0959 | 242.9 |
| 24 | There are $n$ values of $x$ ... | TRUE | 0.334 | 0.333 | 0.333 | P | 1322.7363 | 125.9 |
| 25 | Sixteen chairs are arrange... | FALSE | 0.336 | 0.333 | 0.332 | P | 1322.3235 | 345.3 |
| 26 | Let $S$ be the set of vertice... | TRUE | 0.333 | 0.334 | 0.333 | M | 1321.531 | 165.7 |
| 27 | Let $A_1 A_2 \ldots A_{11}$... | TRUE | 0.333 | 0.334 | 0.333 | M | 1323.1938 | 184.6 |
| 28 | Let the sequence of ratio... | FALSE | 0.335 | 0.333 | 0.332 | P | 1322.7058 | 479.7 |
| 29 | Let $\triangle ABC$ be ... | TRUE | 0.334 | 0.333 | 0.333 | P | 1323.3822 | 219.5 |
| 30 | Let... | FALSE | 0.335 | 0.331 | 0.334 | P | 1323.7086 | 299.4 |

# I  ALGORITHM DETAILS

This appendix provides detailed algorithmic implementations of DNON's key components, presented as pseudocode that could guide practical implementation.

---

**Algorithm 1** DNON Training Procedure

---

**Require:** Training dataset $\mathcal{D} = \{(x_i, y_i)\}_{i=1}^N$, learning rate $\eta$, regularization coefficient $\lambda$
**Ensure:** Trained DNON modules $\{LLM\text{-}P, LLM\text{-}M, LLM\text{-}R, LLM\text{-}E\}$ with parameters $\Theta$
1: Initialize module parameters $\Theta_P, \Theta_M, \Theta_R, \Theta_E$
2: Initialize memory states $S_i^* \leftarrow \arg\min_S \mathcal{L}(S)$ for each module $i$
3: Initialize routing parameters $\omega_{ij} \leftarrow \frac{1}{3}$ for $i \neq j$
4: Initialize Fisher information matrices $G_{\Theta_i} \leftarrow \mathbb{E}\left[\nabla_{\Theta_i}\mathcal{L}\,\nabla_{\Theta_i}\mathcal{L}^\top\right]$
5: **for** each minibatch $(x, y) \in \mathcal{D}$ **do**
6:     **for** each module $i \in \{P, M, R, E\}$ **do**
7:         Compute routed input: $x_i = \sum_j \omega_{ji} T_{j \to i}(S_j)$
8:         Forward propagate: $h_i = LLM\text{-}i(x_i; \Theta_i)$
9:     **end for**
10:    Compute global prediction $\hat{y} = f(h_P, h_M, h_R, h_E)$
11:    Compute loss $\mathcal{L} = \ell(y, \hat{y})$
12:    **for** each module $i$ **do**
13:        Compute natural gradient: $\tilde{\nabla}_{\Theta_i} = G_{\Theta_i}^{-1}\nabla_{\Theta_i}\mathcal{L}$
14:        Update parameters: $\Theta_i \leftarrow \Theta_i - \eta\tilde{\nabla}_{\Theta_i} - \lambda R(\Theta_i)$
15:        Update Fisher matrix: $G_{\Theta_i} \leftarrow (1 - \beta)G_{\Theta_i} + \beta\nabla_{\Theta_i}\mathcal{L}\,\nabla_{\Theta_i}\mathcal{L}^\top$
16:        Update routing weights: $\omega_{ij} \leftarrow \text{softmax}_j(I(S_i; S_j))$
17:    **end for**
18: **end for**

---

**Implementation note.** Algorithm 1 is instantiated in our codebase as follows:

- The overall training loop is implemented in `TrainableDNONSystem_SQuAD.train` within `dnon_system_trainable_squad.py`.

- Router parameter updates correspond to `TrainableDynamicRouter_SQuAD.update_parameters` in `dynamic_router_trainable_squad.py`.

- Dataset splits (train / dev / test) are handled by `split_dataset_squad` in `dataset_utils_squad.py`.

---

**Algorithm 2** Information Transfer Between Modules

---

1: **Input:** Source $S_i$, target $S_j$, manifolds $M_i, M_j$, kernel bandwidth $\sigma$
2: **Output:** Updated $S_j$
3: **Parameters:** $N = 1000$                                                              ▷ Number of samples
4:     $\sigma = 1.0$                                                                        ▷ Kernel bandwidth
5: **Definitions:**
6: Proposal distribution: $q(m) = \mathcal{N}(S_i, \sigma^2 \boldsymbol{I})$
7: Geometric tensor: $G_i = \boldsymbol{I} + 0.1 \cdot \text{Cov}(\text{recent\_states}_i)$
8: Integration function:

$$\phi_j(\text{current}, \text{transfer}) = 0.9 \cdot \text{current} + 0.1 \cdot \exp\big(-\|\text{transfer} - \text{current}\|^2\big) \cdot \text{transfer}$$

9: Initialize accumulator $A = 0$, normalization $Z = 0$
10: **for** $k = 1$ to $N$ **do**
11:     Sample $m_k \sim q(m)$
12:     Proposal density: $q_{\text{val}} = (2\pi\sigma^2)^{-d/2} \cdot \exp\big(-\|m_k - S_i\|^2/(2\sigma^2)\big)$
13:     Kernel: $K = \exp\big(-\frac{1}{2}(m_k - S_i)^T G_i^{-1}(m_k - S_i)/\sigma^2\big)$
14:     Transfer map: $\psi_{ij} = W_{ij} m_k + b_{ij}$                           ▷ $W_{ij}, b_{ij}$ are learned parameters
15:     Importance weight: $w = K/q_{\text{val}}$
16:     Clip weight: $w \leftarrow \text{clip}(w, 1e-6, 1e6)$
17:     Accumulate: $A \leftarrow A + w \cdot \psi_{ij}$, $Z \leftarrow Z + w$
18: **end for**
19: Normalize transfer: $T \leftarrow A/Z$ **if** $Z > 0$ **else** $S_i$
20: Update target: $S_j \leftarrow \phi_j(S_j, T)$
21: **Return** $S_j$

---

**Implementation note.** Algorithm 2 is instantiated in the codebase through the module–communication components listed below:

- `BaseAgent.update_state` in `dnon_base_modules.py`, which applies the manifold projection and state–update operators corresponding to the abstract $\Phi_{i \to j}$ mappings.

- `RiemannianManifold.project` and `RiemannianManifold.geodesic` (same file), which implement the information–geometry transformation used when transferring information between module states.

- `DynamicRouter.compute_routing_matrix` in `dynamic_router_StrategyQA.py`, which provides the MI-based routing weights that determine the contribution of each module during information exchange.

- `MemoryAgent.process_with_memory` in `dnon_modules_squad.py`, which integrates perception and reasoning messages using the same information-transfer principle for the SQuAD experiments.

---

**Algorithm 3** Neural Mutual Information Estimation

---

1: **Input:** Samples from distributions $p(x)$, $p(y)$, joint distribution $p(x, y)$
2: **Output:** Estimated mutual information $I(X; Y)$
3: Initialize neural network $T_\theta$ with parameters $\theta$
4: **for** training iteration $t = 1$ to $T$ **do**
5:     Sample batch $\{(x_i, y_i)\}_{i=1}^B$ from $p(x, y)$
6:     Sample batch $\{x_j\}_{j=1}^B$ from $p(x)$
7:     Sample batch $\{y_k\}_{k=1}^B$ from $p(y)$
8:     **Joint term (corrected):**
9:        JointScore $\leftarrow \frac{1}{B} \sum_{i=1}^B T_\theta(x_i, y_i)$
10:     **Marginal term (numerically stable):**
11:       marginal_scores $\leftarrow [T_\theta(x_j, y_k)$ for all $j, k \in \{1..B\}]$
12:       max_score $\leftarrow \max($marginal_scores$)$
13:       MargScore $\leftarrow$ max_score $+ \log\left(\frac{1}{B^2} \sum \exp(\text{marginal\_scores} - \text{max\_score})\right)$
14:     **MINE Loss:** Loss $\leftarrow -($JointScore $-$ MargScore$)$
15:     Update $\theta$ using gradient descent to minimize Loss
16: **end for**
17: **Final estimate:** $I(X; Y) \leftarrow$ JointScore $-$ MargScore with final $\theta$
18: **Return** $I(X; Y)$

---

**Implementation note.** Algorithm 3 is instantiated in our system using the components listed below:

- `MutualInformationEstimator` in `mi_estimators.py`, which implements histogram-based, JS-divergence–based, and kernel-based MI approximations used during module interaction.

- `MineEstimator` in the same file, providing a neural estimation variant inspired by MINE for higher-dimensional module states.

- `DynamicRouter.compute_routing_matrix` in `dynamic_router_StrategyQA.py`, which consumes the MI estimates to construct the routing weights used in the system.

- `TrainableDynamicRouter_SQuAD` in `dynamic_router_trainable_squad.py`, which integrates neural MI estimates into the learnable routing mechanism for the SQuAD experiments.

---

**Algorithm 4** Dynamic Information Routing

---

**Require:** States $\{S_i(t)\}$; temperature $\beta$; stability parameters
**Ensure:** Updated routing matrix $\Omega(t)$

1: Parameters:
 • $\texttt{min\_mi} = 0.01, \texttt{max\_mi} = 2.0$           // MI bounds for stability
 • $\texttt{min\_weight} = 0.05$              // Minimum routing weight
 • $\alpha = 0.3$                    // Smoothing factor
2: Initialize new routing matrix $\Omega^{new} = \mathbf{0}_{4 \times 4}$
3: **for** each pair $(i, j)$ where $i \neq j$ **do**
4:  Exclude $S_{-ij} = \{S_k \mid k \neq i, j\}$
5:  Estimate MI using Algorithm: $MI = I(S_i; S_j \mid S_{-ij})$
6:  **Stability:** $MI \leftarrow \text{clip}(MI, \texttt{min\_mi}, \texttt{max\_mi})$
7:  Apply temperature scaling: score $= \exp(\beta \cdot MI)$
8:  Store: $\Omega^{new}[i, j] \leftarrow$ score
9: **end for**
10: **Corrected Normalization:** Row normalize (each module's outgoing weights)
11: **for** each row $i$ **do**
12:  $row\_sum \leftarrow \sum_{j \neq i} \Omega^{new}[i, j]$
13:  **for** each $j \neq i$ **do**
14:   $\Omega^{new}[i, j] \leftarrow \Omega^{new}[i, j]/row\_sum$
15:   **Stability:** $\Omega^{new}[i, j] \leftarrow \max(\texttt{min\_weight}, \Omega^{new}[i, j])$
16:  **end for**
17:  $row\_sum \leftarrow \sum_{j \neq i} \Omega^{new}[i, j]$
18:  **for** each $j \neq i$ **do**
19:   $\Omega^{new}[i, j] \leftarrow \Omega^{new}[i, j]/row\_sum$
20:  **end for**
21: **end for**
22: **Smoothing:** $\Omega(t) \leftarrow (1 - \alpha) \cdot \Omega(t - 1) + \alpha \cdot \Omega^{new}$
23: **return** $\Omega(t)$

---

**Implementation note.** The routing mechanism in Algorithm 4 is implemented in our codebase through the following components:

- `DynamicRouter` (file: `dynamic_router_StrategyQA.py`) — core routing logic, including mutual-information computation.

- `compute_routing_matrix` and `compute_routing_matrix_advanced` — routing weight construction.

- `TrainableDynamicRouter_SQuAD` (file: `dnon_trainable_router_squad.py`) — trainable variant used in SQuAD experiments.

---

**Algorithm 5** Three-Tier Memory Operations

---

1: **Input:** Perception state $S_P$, memories $S_M = [S_M^{ST}, S_M^{LT}, S_M^{DS}]$, executive state $S_E$
2: **Output:** Updated memories $(S_M^{ST}, S_M^{LT}, S_M^{DS})$ and memory-to-reasoning weights $(\omega_{ST}, \omega_{LT}, \omega_{DS})$
3: **Parameters:**
4: $\tau_{ST}$: STM decay time constant
5: $\theta_{LT}$: LTM consolidation threshold
6: $\theta_{DS}$: DSM emotional threshold
7: $\eta_{LT}$: LTM learning rate
8: $\alpha_{DS}$: DSM learning rate
9: **Short-term memory (STM) update:**
10: decay $= S_M^{ST}/\tau_{ST}$
11: Define $F_M^{ST}(S_M^{ST}, S_P) = 0.1 \cdot (S_M^{ST} \odot \tanh(S_P))$
12: $\text{noise}_{STM} \sim \mathcal{N}(0, 0.01)$
13: $S_M^{ST} \leftarrow S_M^{ST} + F_M^{ST}(S_M^{ST}, S_P) - \text{decay} + \text{noise}_{STM}$
14: **Long-term memory (LTM) consolidation:**
15: Define importance $\gamma = \text{cosine\_similarity}(S_M^{ST}, \text{current\_task\_embedding})$
16: **if** $\gamma > \theta_{LT}$ **then**
17: $\quad$ Define $\Phi(S_M^{ST}, S_M^{LT}) = \gamma \cdot (S_M^{ST} - 0.9 \cdot S_M^{LT})$
18: $\quad \text{noise}_{LTM} \sim \mathcal{N}(0, 0.001)$
19: $\quad S_M^{LT} \leftarrow S_M^{LT} + \eta_{LT} \cdot \Phi(S_M^{ST}, S_M^{LT}) + \text{noise}_{LTM}$
20: **end if**
21: **Deep subconscious memory (DSM) update:**
22: Define $E(S_E) = \sigma(S_E^\top w_{\text{emotional}})$
23: Emotional salience $\epsilon = E(S_E)$
24: **if** $\epsilon > \theta_{DS}$ **then**
25: $\quad$ Define $\Psi(S_M^{LT}, S_M^{DS}, S_E) = 0.4 \cdot (S_M^{LT} - S_M^{DS}) + 0.6 \cdot \epsilon \cdot S_E$
26: $\quad S_M^{DS} \leftarrow S_M^{DS} + \alpha_{DS} \cdot \Psi(S_M^{LT}, S_M^{DS}, S_E)$
27: **end if**
28: **Memory-to-reasoning routing weights:**
29: Compute mutual information scores:
30: $I_{STM} = I(S_M^{ST}; \text{current\_reasoning\_context})$
31: $I_{LTM} = I(S_M^{LT}; \text{current\_reasoning\_context})$
32: $I_{DSM} = I(S_M^{DS}; \text{current\_reasoning\_context})$
33: Temperature scaling: $scores = [\exp(\lambda I_{STM}), \exp(\lambda I_{LTM}), \exp(\lambda I_{DSM})]$
34: Softmax normalization: $[\omega_{ST}, \omega_{LT}, \omega_{DS}] = \text{softmax}(scores)$
35: Apply minimum weights: $\omega_i = \max(0.1, \omega_i), \ i \in \{ST, LT, DS\}$
36: Re-normalize: $[\omega_{ST}, \omega_{LT}, \omega_{DS}] \leftarrow [\omega_{ST}, \omega_{LT}, \omega_{DS}]/(\omega_{ST} + \omega_{LT} + \omega_{DS})$
37: **Return:** $(S_M^{ST}, S_M^{LT}, S_M^{DS}), (\omega_{ST}, \omega_{LT}, \omega_{DS})$

---

**Implementation note.** Algorithm 5 is instantiated in our codebase as follows:

- The three memory systems are implemented by `ShortTermMemory`, `LongTermMemory`, and `DeepSubconsciousMemory` in `memory_dynamics_StrategyQA.py`.

- The integration of these components in the QA pipeline is handled by `MemoryAgent.process_with_memory` in `dnon_modules_squad.py`, which coordinates STM retrieval, LTM consolidation, and DSM updates.

- The scalar modulation used in the DSM update is produced by `MemoryAgent.calculate_emotional_salience`, providing the practical instantiation of the abstract emotional term in the DSM dynamics.

---

**Algorithm 6** Convergence Monitoring

---

1: **Input:** States $\{S_i(t)\}$, convergence threshold $\epsilon = 1e-3$
2: **Output:** Lyapunov value $V$, convergence status
3: **Parameters:**
4: $\alpha = 0.1$, $\beta = 0.01$, noise_bound $= 0.1$
5: $\Delta t$                  ▷ Time step for derivative approximation
6: **Optimal State Estimation:**
7: **if** first iteration **then**
8:      $S_i^* \leftarrow S_i(t)$                  ▷ Initialize optimal states
9: **else**
10:      $S_i^* \leftarrow 0.99 \cdot S_i^* + 0.01 \cdot S_i(t)$        ▷ Exponential moving average update
11: **end if**
12: Compute Lyapunov value: $V(t) \leftarrow \sum_i \|S_i(t) - S_i^*\|^2$
13: Determine convergence status:
14: **if** $V(t-1)$ exists **then**
15:      $dV/dt \leftarrow (V(t) - V(t-1))/\Delta t$
16:      **if** $dV/dt \leq -\alpha \cdot V(t) + \beta \cdot$ noise_bound **then**
17:          status $\leftarrow$ "Converging"
18:      **else**
19:          status $\leftarrow$ "Not Converging"
20:      **end if**
21: **else**
22:      status $\leftarrow$ "Initializing"
23: **end if**
24: **if** $V(t) < \epsilon$ **then**
25:      status $\leftarrow$ "Converged"
26: **end if**
27: Store $V(t)$ for next iteration
28: **Return** $(V, \text{status})$

---

**Implementation note.** Algorithm 6 is instantiated in our system through the following components:

- `ConvergenceAnalyzer` in `convergence_analysis.py`, which implements the Lyapunov-style stability metric, sliding-window monitoring, and associated convergence checks for module states.

- `compute_lyapunov_function` and `check_convergence` within the same class, corresponding to the stability evaluation and threshold-based detection used during system execution.

- `AdaptiveConvergenceAnalyzer_SQuAD` in `dnon_convergence_squad.py`, which provides a task-specific variant of the convergence monitoring logic for the SQuAD experiments.

## J    DEFAULT NOTATION

### J.1    NUMBERS AND ARRAYS

| | |
|---|---|
| $a$ | A scalar (integer or real) |
| $\boldsymbol{a}$ | A vector |
| $\boldsymbol{A}$ | A matrix |
| $\mathbf{A}$ | A tensor |
| $\boldsymbol{I}_n$ | Identity matrix with $n$ rows and $n$ columns |
| $\boldsymbol{I}$ | Identity matrix (dimensionality implied) |
| $e^{(i)}$ | Standard basis vector with 1 at position $i$ |
| $\mathrm{diag}(\boldsymbol{a})$ | Diagonal matrix with entries $\boldsymbol{a}$ |
| $\mathrm{a}$ | Scalar random variable |
| $\mathbf{a}$ | Vector random variable |
| $\mathbf{A}$ | Matrix random variable |
| $\Omega(t)$ | Routing matrix at time $t$ |
| $\Omega^{new}$ | Newly computed routing matrix before smoothing |

### J.2    SETS AND GRAPHS

| | |
|---|---|
| $\mathbb{A}$ | A set |
| $\mathbb{R}$ | The set of real numbers |
| $\{0, 1\}$ | The set containing 0 and 1 |
| $\{0, 1, \ldots, n\}$ | Set of integers from 0 to $n$ |
| $[a, b]$ | Closed interval from $a$ to $b$ |
| $(a, b]$ | Half-open interval $(a, b]$ |
| $\mathbb{A} \backslash \mathbb{B}$ | Set subtraction |
| $\mathcal{G}$ | A graph |
| $Pa_{\mathcal{G}}(\mathrm{x}_i)$ | Parents of $\mathrm{x}_i$ in graph $\mathcal{G}$ |

### J.3    INDEXING

| | |
|---|---|
| $a_i$ | Element $i$ of vector $\boldsymbol{a}$ |
| $a_{-i}$ | All elements except $i$ |
| $A_{i,j}$ | Element $(i, j)$ of matrix $\boldsymbol{A}$ |
| $\boldsymbol{A}_{i,:}$ | Row $i$ of $\boldsymbol{A}$ |
| $\boldsymbol{A}_{:,i}$ | Column $i$ of $\boldsymbol{A}$ |
| $A_{i,j,k}$ | Element $(i, j, k)$ of tensor $\mathbf{A}$ |
| $\mathbf{A}_{:,:,i}$ | 2-D slice of 3-D tensor |
| $\mathrm{a}_i$ | Element $i$ of random vector $\mathbf{a}$ |

## J.4 CALCULUS

| | |
|---|---|
| $\dfrac{dy}{dx}$ | Derivative of $y$ wrt $x$ |
| $\dfrac{\partial y}{\partial x}$ | Partial derivative |
| $\nabla_{\boldsymbol{x}} y$ | Gradient wrt vector $\boldsymbol{x}$ |
| $\nabla_{\boldsymbol{X}} y$ | Gradient wrt matrix $\boldsymbol{X}$ |
| $\nabla_{\mathsf{X}} y$ | Gradient wrt tensor $\mathsf{X}$ |
| $\dfrac{\partial f}{\partial \boldsymbol{x}}$ | Jacobian of $f : \mathbb{R}^n \to \mathbb{R}^m$ |
| $\nabla_{\boldsymbol{x}}^2 f(\boldsymbol{x})$ or $\boldsymbol{H}(f)(\boldsymbol{x})$ | Hessian of $f$ |
| $\displaystyle\int f(\boldsymbol{x})d\boldsymbol{x}$ | Integral over entire domain |
| $\displaystyle\int_{\mathbb{S}} f(\boldsymbol{x})d\boldsymbol{x}$ | Integral over set $\mathbb{S}$ |

## J.5 PROBABILITY AND INFORMATION THEORY

| | |
|---|---|
| $P(\mathrm{a})$ | Probability distribution over discrete variable |
| $p(\mathrm{a})$ | Probability distribution over continuous variable |
| $\mathrm{a} \sim P$ | a has distribution $P$ |
| $\mathbb{E}_{\mathrm{x} \sim P}[f(x)]$ | Expectation |
| $\mathrm{Var}(f(x))$ | Variance |
| $\mathrm{Cov}(f(x), g(x))$ | Covariance |
| $H(\mathrm{x})$ | Shannon entropy |
| $D_{\mathrm{KL}}(P\|Q)$ | Kullback-Leibler divergence |
| $\mathcal{N}(\boldsymbol{x}; \boldsymbol{\mu}, \boldsymbol{\Sigma})$ | Gaussian distribution |
| $I(X; Y)$ | Mutual information between $X$ and $Y$ |
| $I(X; Y|Z)$ | Conditional mutual information |
| $q(m)$ | Proposal distribution for sampling manifold elements in transfer |

## J.6 FUNCTIONS

| | |
|---|---|
| $f : \mathbb{A} \to \mathbb{B}$ | Function with domain $\mathbb{A}$ and codomain $\mathbb{B}$ |
| $f \circ g$ | Composition: $(f \circ g)(x) = f(g(x))$ |
| $f(\boldsymbol{x}; \boldsymbol{\theta})$ | Function of $\boldsymbol{x}$ parametrized by $\boldsymbol{\theta}$ |
| $\log x$ | Natural logarithm |
| $\sigma(x)$ | Logistic sigmoid |
| $\mathrm{softmax}(\boldsymbol{z})_i$ | Softmax function |
| $\phi_j(\mathrm{current}, \mathrm{transfer})$ | Integration function for updating target module state |

## J.7 ITEMIZED DEFAULT NOTATIONS

- Scalars: $a, b, c, \ldots$
- Vectors: $\boldsymbol{a}, \boldsymbol{b}, \boldsymbol{x}, \boldsymbol{y} \in \mathbb{R}^n$
- Matrices: $\boldsymbol{A}, \boldsymbol{B} \in \mathbb{R}^{m \times n}$

- Tensors: $\mathbf{A} \in \mathbb{R}^{d_1 \times \cdots \times d_k}$

- Sets: $\mathbb{A}, \mathbb{B}, \mathbb{D}, \ldots$

- Functions: $f, g, h : \mathbb{R}^n \to \mathbb{R}^m$

- Manifolds: $M_i \subset \mathbb{R}^{d_i}$

- Probability distributions: $p(x), q(x)$

- Expectation: $\mathbb{E}_{x \sim p}[f(x)]$

- Mutual information: $I(X;Y)$, Conditional MI: $I(X;Y|Z)$

- Noise/perturbations: $\xi, \epsilon \sim \mathcal{N}(0, \sigma^2)$

- Learning rates: $\eta, \alpha$

- Kronecker product: $\otimes$

- Step size for discretization: $\Delta t$

- Contraction parameters: $\lambda_i > 0$

- Routing parameters: $\beta > 0$

- Importance weight for transfer: $w$

- Normalized transfer output: $T$

## J.8 MEMORY AND COMMUNICATION

| | |
|---|---|
| $S_M^{ST}, S_M^{LT}, S_M^{DS}$ | Short-term, Long-term, and Deep Subconscious memory module states |
| $S_M^{unified}, S_M^{external}$ | Alternative memory architectures for theoretical ablation |
| $X_{recent}$ | Recent inputs for short-term memory |
| $X_{distinct}$ | Distinct inputs for long-term memory |
| $X_{emotional}$ | Emotional/procedural inputs for deep subconscious memory |
| $X_{diverse}$ | Diverse inputs for external memory or ablations |
| $T_{i \to j}$ | Information Transfer Operator from module $i$ to $j$ (kernel-based) |
| $T_{i \to j}^{direct}$ | Direct state transfer operator (simplified ablation) |
| $T_{i \to j}^{att}$ | Attention-based transfer operator without manifold awareness |
| $K(S_i, m)$ | Kernel function over source module state $S_i$ and manifold element $m$ |
| $\psi_{i \to j}(m)$ | Module-specific transfer function mapping $m$ to destination module $j$ |
| $W_{ij}, b_{ij}$ | Learned linear projection weights for transfer map |
| $S_j^*$ | Target state in module $j$ manifold for transfer operator comparison |

## J.9 THREE-TIER MEMORY NOTATIONS

| | |
|---|---|
| $F_M^{ST}(S_M^{ST}, S_P)$ | Short-term memory update function: $0.1 \cdot (S_M^{ST} \odot \tanh(S_P))$ |
| $\Phi(S_M^{ST}, S_M^{LT})$ | LTM novelty-weighted consolidation function: $\gamma \cdot (S_M^{ST} - 0.9 \cdot S_M^{LT})$ |
| $\Psi(S_M^{LT}, S_M^{DS}, S_E)$ | DSM update function: $0.4 \cdot (S_M^{LT} - S_M^{DS}) + 0.6 \cdot \epsilon \cdot S_E$ |
| $\gamma$ | LTM importance score (cosine similarity between STM and current task embedding) |
| $\epsilon$ | Emotional salience: $\sigma(S_E^T w_{\text{emotional}})$ |
| $w_{\text{emotional}}$ | Learned weight vector for computing emotional salience |
| $I_{STM}, I_{LTM}, I_{DSM}$ | Mutual information between STM, LTM, DSM and reasoning context |
| $\lambda$ | Temperature parameter for softmax scaling of MI scores |
| $\omega_{ST}, \omega_{LT}, \omega_{DS}$ | Memory-to-reasoning transfer weights for STM, LTM, DSM |
| $\xi_M^{ST}, \xi_M^{LT}$ | Gaussian noise added to STM and LTM updates |

## J.10 CONVERGENCE AND MONITORING NOTATIONS

| | |
|---|---|
| $V(t)$ | Lyapunov value at time $t$ |
| $S_i^*$ | Optimal state of module $i$ for monitoring |
| $dV/dt$ | Time derivative of Lyapunov function |
| $\epsilon$ | Convergence threshold |
| $\alpha, \beta$ | Contraction and noise parameters for monitoring |
| status | Convergence status label ("Converged", "Initializing", etc.) |
| $\Delta t$ | Time step for derivative approximation |

