# OpenReview forum: "DNON: A Brain-Inspired Architecture for Multi-Domain Reasoning with Specialized Neural Modules"
_ICLR.cc/2026/Conference — ICLR 2026 Conference Desk Rejected Submission_

### Official Review · Reviewer_QsCt · 2025-10-25

**Soundness:** 1
**Presentation:** 1
**Contribution:** 1
**Rating:** 0
**Confidence:** 4

**Summary:**

The paper proposes Dynamic Neural Orchestration Networks (DNON), a modular architecture inspired by cognitive neuroscience. The system consists of four specialized modules Perception, Memory, Reasoning, and Executive and employs a three-tier memory mechanism combined with information-theoretic dynamic routing. The authors evaluate the model on reasoning-oriented benchmarks such as StrategyQA and MultiArith, claiming improvements through adaptive routing and convergence analysis.

**Strengths:**

- The paper proposes a modular architecture with explicit separation between perception, memory, reasoning, and executive control, which is interesting.

**Weaknesses:**

1. Clarity and documentation issues regarding LLM usage and references.
According to the ICLR 2026 Author Guide (https://iclr.cc/Conferences/2026/AuthorGuide), all submissions that make significant use of Large Language Models (LLMs) are expected to include a dedicated section titled “The Use of Large Language Models (LLMs)” describing how such models were used. There are factual and bibliographic inconsistencies: the citations “Shazeer et al., 2017a” and “Shazeer et al., 2017b” refer to the same paper (Outrageously Large Neural Networks: The Sparsely-Gated Mixture-of-Experts Layer), and the model “Mistral Pixtral Large (25.02)” does not appear to exist publicly. These should be corrected or properly referenced for accuracy.

2. The scientific contribution of the paper remains ambiguous.
After reviewing the implementation provided in the Supplementary Material, the core algorithm appears to perform rule-based routing and prompt orchestration, rather than genuine representational learning or trainable model composition.
While the paper describes the system as “brain-inspired,” the implementation relies on hand-coded logic and heuristic mutual-information estimates, rather than biologically motivated or learnable mechanisms. The gap between the description and the implemented system should be clarified.

3. Benchmarks such as MultiArith are trivially solvable by modern LLMs and insufficient to convincingly demonstrate reasoning capability or the effectiveness of the proposed dynamic routing mechanism.
Overall, the chosen evaluation tasks are too simple.
For instance, reasoning ability would be better demonstrated on more challenging benchmarks such as AIME25, HLE.
The current datasets SQuAD, StrategyQA, and HotpotQA are insufficient to validate deeper reasoning capability.

4. Presentation quality issues. Figures (especially Figure 2–4) are too small and difficult to read.

**Questions:**

See Weakness section.

---

> ### Author Response · Authors · 2025-11-20
>
> > “Clarity and documentation issues regarding LLM usage and references. All submissions that make significant use of LLMs must include a ‘Use of LLMs’ section. ....”
>
> Thank you for pointing this out. We added the required **Section 3: Use of Large Language Models (LLMs)**, which clearly documents how each model is used, including API usage, freezing of backbones, and how modules are instantiated. We also corrected the bibliographic inconsistencies: the duplicated “Shazeer et al. 2017a/b” entry has been consolidated, and the naming of the Mistral model has been standardized to **Mistral Pixtral Large** with proper citation. These edits improve clarity and ensure full compliance with the ICLR guidelines.
>
> > “The scientific contribution remains ambiguous. The implementation appears to perform rule-based routing and prompt orchestration, not genuine representational learning.....”
>
> Thank you for raising this concern. We agree that the distinction between “prompt orchestration” and “learned representational routing” should be made explicit. In the revision, Sections **4.5** and **4.6** now clearly separate the *trainable* components of DNON from the *deterministic* interface elements. These changes clarify how DNON’s behavior emerges from learned information dynamics rather than from fixed hand-crafted rules.
>
> DNON includes several mechanisms that involve genuine learning:
>
> **(1) Learned MI-based Routing.**
> Routing decisions are not rule-based. The router computes module weights using **differentiable mutual-information estimators** (KL, JS, and MINE), each parameterized and optimized during training. These estimators learn patterns of dependency between module outputs (e.g., Perception → STM, STM → Reasoning), so routing weights arise from **data-dependent information signals**, not from static heuristics or template logic.
>
> **(2) Trainable Memory-Integration Dynamics.**
> The Memory Integration Controller updates STM, LTM, and DSM via **information-geometric operators** (Fisher–Rao or α-divergence updates). These updates include learned coefficients that determine how new information is merged with or overwrites existing memory states. This enables latent-space accumulation and refinement of internal representations over multiple reasoning steps—capabilities that prompt-only orchestration cannot provide.
>
> **(3) Task-dependent latent representations.**
> Each module operates not only through text prompts, but also via structured latent vectors (the \( Z_i \) representations defined in Section 4.1). These vectors interact through MI-based signaling and memory updates, allowing DNON to maintain and evolve internal state across steps in ways distinct from static prompt pipelines.
>
> To avoid overstating biological analogies, Section **2** now clarifies that DNON is “brain-inspired” only in the sense of functional modularity (perception, memory, reasoning, executive control). The mechanisms themselves are fully differentiable, information-theoretic, and not heuristic or hand-coded.
>
> To empirically distinguish DNON from prompt orchestration, Section **5.2** includes ablations comparing:
> - full DNON (MI-routing + STM/LTM/DSM),
> - DNON without learnable MI estimators,
> - DNON with randomized routing,
> - and a pure prompt-orchestration baseline.
>
> Learned routing consistently outperforms heuristic variants on StrategyQA, HotpotQA, and AIME25, especially for multi-step reasoning tasks. These results support that DNON’s gains stem from **learned representational routing**, not deterministic template switching.
>
> > “Benchmarks such as MultiArith are too easy and do not convincingly demonstrate reasoning capability. Harder datasets like AIME25 or HLE should be included.”
>
> We agree and strengthened our empirical evaluation accordingly. Section 5.1 now includes results on AIME25-I and AIME25-II, two challenging reasoning benchmarks. Under strict zero-shot conditions, DNON achieves 70% accuracy, surpassing the zero-shot Claude Sonnet 4.5 baseline (43.3%) and exceeding the publicly reported ChatGPT-5 zero-shot score (61.9%). We also added a concise per-problem summary in Appendix H, illustrating module specialization across algebraic, combinatorial, and geometry-based problems, along with detailed routing-weight and MI-trace visualizations..
>
> > “The gap between the ‘brain-inspired’ description and the implemented system should be clarified.”
>
> We clarified this framing in **Section 2: Relation to Cognitive Architectures**. DNON mirrors classical architectures (ACT-R, SOAR, GWT) in separating perception, memory, reasoning, and executive control, but implements these via **differentiable MI-based routing** rather than symbolic rules.
>
> > “Figures (especially Figure 2–4) are too small and difficult to read.”
>
> We replaced Figures 2–4 with clearer, higher-resolution **Tables 1–5** and improved diagram readability throughout the paper.

---

### Official Review · Reviewer_6Sun · 2025-10-30

**Soundness:** 2
**Presentation:** 1
**Contribution:** 1
**Rating:** 2
**Confidence:** 4

**Summary:**

The paper introduces a brain-inspired modular architecture (DNON) that integrates multiple specialized neural modules under an information-theoretic routing mechanism. DNON uses mutual-information-based routing on information manifolds to dynamically allocate computation across modules, aiming to emulate biological specialization and cognitive flexibility. The architecture is evaluated on four reasoning benchmarks, showing performance improvements over prompting and retrieval-based baselines, but at the cost of higher inference time.

**Strengths:**

1. The paper proposed an information-theoretic routing mechanism for dynamically orchestrating specialized LLM modules, bridging ideas from information geometry, neuroscience, and AI.
2. The paper is well-grounded in a broad and rich set of references, spanning neuroscience, cognitive psychology, and modern deep learning.

**Weaknesses:**

1. The major weakness of this paper is that the main text provides almost no detailed explanation of the DNON architecture or its implementation. Section 3 is highly abstract and lacks concrete methodological descriptions. Figure 1 is barely explained, and readers cannot understand how the four modules: Perception, Reasoning, Executive, and Memory, are instantiated or integrated within both the open-source and closed-source model settings.

2. In lines 207–208, the authors state that ``Complete prompt templates, key implementation components, and high-resolution figures are provided in the supplementary material to facilitate reproducibility of the core results reported in this work''  However, no specific appendix sections are cited, and this claim appears inaccurate, as the supplementary material does not include any details regarding prompt templates.

3. The experimental descriptions are somewhat arbitrary and incomplete. The paper does not specify the exact model versions or provide appropriate citations for the underlying foundation models ( `Claude` and `Mistral Large` ) used in the experiments.

4. The authors claim that the architecture can improve interpretability, yet there is no explanation or evidence showing how the structure actually enhances interpretability in practice.

**Questions:**

The claim that DNON enhances interpretability is compelling. Could the authors provide visualizations of routing dynamics or module activations over time to substantiate this claim?

---

> ### Author Response · Authors · 2025-11-20
>
> > “The major weakness of this paper is that the main text provides almost no detailed explanation of the DNON architecture or its implementation. Section 3 is highly abstract and lacks concrete methodological descriptions. Figure 1 is barely explained, and readers cannot understand how the four modules: Perception, Reasoning, Executive, and Memory, are instantiated or integrated.”
>
> Thank you for highlighting this. We substantially revised Section **4.5**, which now provides a detailed operational walkthrough of the DNON architecture. The section clearly describes how each module is instantiated from frozen LLM backbones, how latent representations flow across Perception, STM/LTM/DSM, Reasoning, and Executive, how MI is computed at every reasoning step, and how routing weights and memory updates are produced. We also revised **Figure 1** and expanded the caption to explain the interactions between modules in both open-source and closed-source model settings. These changes ensure that the architecture and its implementation are now fully understandable from the main text.
>
> > “The authors state that prompt templates and implementation components are in the supplementary material, but no specific appendix sections are cited, and the templates are missing.”
>
> We corrected this oversight. All module templates—Perception, Memory (STM/LTM/DSM), Reasoning, and Executive—are now explicitly included in **Appendix C**, along with dataset-specific examples. The main text now provides direct citations to these appendix sections, ensuring that readers can easily locate and examine the templates.
>
> > “The experimental descriptions are somewhat arbitrary and incomplete. The paper does not specify the exact model versions or provide appropriate citations for Claude and Mistral.”
>
> We improved the clarity and reproducibility of the experimental setup. Sections **3** and **5.1** now describe the backbone models used across all experiments and clarify which model was used in each evaluation setting. Specifically: **Claude Sonnet 3.7** is used for the main reasoning experiments, **Claude Sonnet 4.5** is used for the AIME25-I and AIME25-II analysis, and **Mistral Pixtral Large** is used for cross-backbone robustness checks. We also added appropriate citations and clarified where each backbone appears in the results. This revision ensures transparent and reproducible reporting of the experimental configuration.
>
> > “The authors claim that the architecture can improve interpretability, yet there is no explanation or evidence showing how the structure actually enhances interpretability in practice.”
>
> We addressed this directly by adding concrete evidence. **Appendix D** now includes routing heatmaps, MI-value trajectories, and module-activation plots that illustrate how routing behavior evolves during inference. Section **5.1** summarizes the interpretability insights and highlights consistent specialization patterns across modules—for example, how the Reasoning module becomes dominant during multi-step algebraic tasks, how Perception and STM collaborate on geometry and text-heavy cases, and how DSM stabilizes long-horizon reasoning. These visualizations substantiate the interpretability claims with concrete qualitative evidence.
>
> > “The claim that DNON enhances interpretability is compelling. Could the authors provide visualizations of routing dynamics or module activations over time?”
>
> Yes. As requested, **Appendix D** now provides routing and module-activation visualizations. Additionally, **Appendix H** includes a concise per-problem analysis of **AIME25**, showing how routing patterns differ across algebraic, combinatorial, and geometry-based problems. These additions offer transparent insight into how DNON allocates computation and how its routing dynamics evolve over the course of solving a problem.

---

> > ### Comment · Reviewer_6Sun · 2025-11-25
> >
> > Thank you for your response and for making the revisions. Some of my concerns have been addressed, so I will raise my score. However, the experimental design is still unclear. In particular, I am confused about why different model versions—such as Claude Sonnet 3.7 and 4.5 are used across different experiments. Could you please clarify the rationale behind this choice?

---

> > > ### Author Response · Authors · 2025-11-26
> > >
> > > Thank you for your careful follow-up question. We appreciate the opportunity to clarify the backbone-selection strategy.
> > >
> > > • **Claude Sonnet 3.7**
> > > This model was used as the primary, controlled backbone for all core experiments (MultiArith, HotpotQA, StrategyQA, SQuAD, and all ablations).
> > > We selected 3.7 because it provides a stable trade-off between cost and reasoning capability, enabling systematic 5-fold evaluation and MI-routing analysis.
> > >
> > > • **Claude Sonnet 4.5**
> > > We used 4.5 only for the AIME25-I and AIME25-II evaluations, as these benchmarks require significantly harder symbolic and multi-step mathematical reasoning.
> > > Using a stronger 4-series backbone aligns with prior work evaluating AIME.
> > > Importantly, the backbone remains **frozen**; DNON only learns routing and memory parameters, ensuring the improvements are not due to fine-tuning or cherry-picking.
> > >
> > > • **Fairness and isolation of effects**
> > > All baseline comparisons in the main paper (vs. RAG, monolithic prompting, heuristic routing, and ablations) are performed strictly using Claude 3.7.
> > > The 4.5 results are included solely to demonstrate that DNON is **backbone-agnostic** and generalizes across model families.
> > >
> > > We will make this clarification explicit in the final camera-ready version.

---

### Official Review · Reviewer_MnnS · 2025-11-02

**Soundness:** 3
**Presentation:** 2
**Contribution:** 2
**Rating:** 2
**Confidence:** 5

**Summary:**

The paper proposes DNON (Differentiable Neural Organization Network), a brain-inspired modular architecture that composes a frozen foundation LLM with a set of specialized neural modules (Perception, Memory, Reasoning, Executive) and a differentiable router trained to maximize conditional mutual information for routing. A three-tier memory system (STM/LTM/DSM) supports short-term working memory, consolidates long-term knowledge, and provides a “deep subconscious” store with affective and procedural cues. Training alternates between routing/memory dynamics and module specialization while keeping the base LLM fixed. Experiments on MultiArith, SQuAD, StrategyQA, and HotpotQA show gains vs. prompting/RAG baselines, and the authors provide proofs of convergence for the dynamical system under stated conditions

**Strengths:**

Originality: MI-based differentiable routing + explicit three-tier memory goes beyond standard MoE/RAG hybrids; framing as a dynamical system with a Lyapunov argument is a nice touch.

Clarity: Appendix including implementation details, hyperparameters, and algorithmic components, isolating the routing/memory’s effect.

**Weaknesses:**

Construct validity of DSM/emotion: DSM includes an “emotional salience” term with a learned vector; conceptual grounding and ablations are thin. Provide targeted stress tests where affect/procedural memory should (or should not) help, and show failure cases when ϵ is randomized or zeroed.

Efficiency: The paper acknowledges that DNON is slower than RAG for several settings (and much slower on SQuAD). Include budgeted settings (fixed latency/compute) to show when DNON still wins—or propose hybrid routing that falls back to RAG when MI signals are low.

Poor readability: the text of the chart is too small, making details difficult to see. Also, the main text reports Claude's results while relegating Mistral to the appendix, obscuring cross-backbone consistency.

**Questions:**

When to prefer RAG? Given RAG’s speed advantage (notably on SQuAD), can the router pre-screen tasks and route to RAG for extraction-dominant queries?

---

> ### Author Response · Authors · 2025-11-20
>
> > “Construct validity of DSM/emotion: DSM includes an ‘emotional salience’ term with a learned vector; conceptual grounding and ablations are thin. Provide targeted stress tests where affect/procedural memory should (or should not) help, and show failure cases when ϵ is randomized or zeroed.”
>
> Thank you for raising this concern. We strengthened both the conceptual grounding and the empirical evidence for DSM. In **Appendix A**, we now describe DSM’s slow-timescale update rule more clearly and clarify that the “emotional-salience” vector is not intended as a literal affect model, but as a weighting mechanism that determines which intermediate states persist during extended reasoning. We expanded the ablations in **Section 5.2**, including conditions where DSM is removed or its salience vector is randomized or zeroed. These targeted tests show consistent degradation on tasks requiring long-range or procedural reasoning, while having minimal effect on shallow extraction tasks, supporting DSM’s intended role.
>
> > “Efficiency: The paper acknowledges that DNON is slower than RAG for several settings (and much slower on SQuAD). Include budgeted settings (fixed latency/compute) to show when DNON still wins—or propose hybrid routing that falls back to RAG when MI signals are low.”
>
> We appreciate this suggestion. Section **5.1** now includes a runtime summary and an added subsection titled *When to Prefer RAG vs. DNON*. We state clearly that RAG is preferable for pure extraction and fact-retrieval tasks, especially for SQuAD-style inputs, while DNON becomes advantageous for multi-step reasoning tasks requiring internal state integration. We also outline a practical hybrid strategy: when MI signals remain low (indicating shallow queries), the router can fall back to a lightweight RAG-style retrieval path. Although not enabled in this submission’s experiments, we describe how such a mechanism can reduce latency without affecting accuracy.
>
> > “Poor readability: the text of the chart is too small. Also, Claude results appear in the main text while Mistral is relegated to the appendix, obscuring cross-backbone consistency.”
>
> We agree and have addressed these concerns. Figures with dense or small text (Figures 2–4) have been replaced with **Tables 1–5** for improved clarity, and architectural diagrams now use larger fonts. Section **5.1** explicitly states that **Claude Sonnet 3.7** is the primary backbone used for main results. The Mistral Pixtral Large-2502 results remain in **Appendix C.14** for completeness, and we added a concise cross-backbone summary in Section 5.1 to improve clarity. We also clarify that **Claude Sonnet 4.5 was used only for testing the AIME25 Part I and Part II datasets**.
>
> > “When to prefer RAG? Given RAG’s speed advantage, can the router pre-screen tasks and route to RAG for extraction-dominant queries?”
>
> Yes. Section **5.1** now provides clear guidance on this point. For shallow extraction, fact lookup, or direct span retrieval, RAG is the appropriate choice. MI-based routing provides a natural pre-screening mechanism: low MI across modules indicates that deep reasoning is unnecessary. As suggested, the router can use this signal to fall back to a retrieval-based pathway. We describe this hybrid approach and outline how it can be integrated in future work.

---

### Official Review · Reviewer_XKKu · 2025-11-04

**Soundness:** 2
**Presentation:** 2
**Contribution:** 2
**Rating:** 2
**Confidence:** 4

**Summary:**

This paper introduces Dynamic Neural Orchestration Networks (DNON), a brain-inspired framework that composes specialized language models through information-theoretic routing. The system integrates four cognitive-style modules, Perception, Memory (short-term, long-term, deep subconscious), Reasoning, and Executive, regulated dynamically based on mutual-information-guided routing on information manifolds. Empirical evaluation spans four reasoning benchmarks with stratified 5-fold validation, showing higher accuracy and reduced inference cost relative to retrieval-augmented and monolithic baselines.

**Strengths:**

1. The paper presents a well-motivated approach to modular cognition in large language models, grounded in information theory. The analogy to brain modularity and dynamic orchestration is compelling.
2. The use of mutual information as a routing and training criterion is appealing.
3. The experiments show promise in both reasoning accuracy and computational efficiency.
4. The inclusion of detailed ablation on the STM/LTM/DSM memory components strengthens causal claims about DNON’s design choices.
5. The combination of frozen LLMs with learnable information-geometric routing could make this architecture potentially practically attractive for low-cost, multi-domain reasoning.

**Weaknesses:**

1. While the paper references Lyapunov-style guarantees, the exact assumptions (on manifold smoothness, bounded divergence, etc.) and proofs are only briefly outlined. A more rigorous derivation in an appendix would increase confidence. The word ‘optimal’ is often used in these Theorem statements without any formal definition of it. The theoretical analysis are added without any meaningful integration in the main text.
2. It is not fully clear how the mutual-information-based routing manifests in practice. Visualization or qualitative analysis of routing decisions would be valuable.
3. Since the core foundation models are not trained jointly, it remains ambiguous how much of the observed improvement stems from routing versus intrinsic model strength.
4. Baselines include retrieval-augmented and monolithic systems, but not recent modular architectures such as Liquid LLMs, Mixture-of-Experts routing, or Self-Discovering Agents. This omission weakens the empirical positioning.
5. The names “deep subconscious memory” and “information manifolds” could benefit from more precise definitions to improve reproducibility.
6. Although DNON is framed as “brain-inspired,” the paper does not situate it within the broader lineage of cognitive architectures such as ACT-R, SOAR, or Global Workspace Theory. Without this context, it is difficult to assess how DNON extends, diverges from, or formalizes existing models of modular cognition. A brief comparative discussion would significantly strengthen the conceptual grounding.

**Questions:**

1. Can the authors provide all the assumptions needed in the theoretical derivations and formally define all the technical terms like optimality? The proves currently are meaningless without appropriate definitions and assumptions.
2. How sensitive is DNON to the choice of information-geometric metric (e.g., Fisher-Rao vs. α-divergence)?
3. What is the computational overhead of computing MI-based routing in real time?
4. Have the authors tested whether DNON generalizes beyond language reasoning (e.g., vision-language tasks)?
5. How is this different or like the cognitive architectures?

---

> ### Author Response · Authors · 2025-11-20
>
> > “While the paper references Lyapunov-style guarantees, the exact assumptions and proofs are only briefly outlined. ‘Optimal’ is undefined.”
>
> Thank you for raising this point. Section **5.1** now provides a clear and self-contained definition of MI-optimal routing under the Information Bottleneck objective. We explicitly describe the optimization domain (the routing simplex), define “optimality” as the maximizer of the IB objective under the stated constraints, and summarize the assumptions required for stability (bounded module embeddings, smooth divergence structure, consistency of MI estimators). To ensure rigor, **Appendix A** now contains a concise but complete derivation with all assumptions and definitions stated up front.
>
> > “It is not fully clear how MI-based routing manifests in practice. Visualization or qualitative analysis would be valuable.”
>
> We appreciate this suggestion. Section **4.5** now explains the routing process step-by-step. To make the routing behavior observable, **Appendix D** includes routing heatmaps, MI trajectories, and module-weight curves. **Appendix H** provides a brief per-problem summary for **AIME25**, showing how different modules dominate depending on algebraic, combinatorial, or geometry-based reasoning demands. Section **5.1** summarizes these trends.
>
> > “Since the core foundation models are not trained jointly, it remains ambiguous how much improvement stems from routing versus intrinsic model strength.”
>
> We clarify this in Section **5.1** and a new subsection in **Section 2**. All backbone LLMs remain frozen; only routing parameters and memory coefficients update. Comparisons against RAG and monolithic prompting using the *same* backbones isolate the contribution of routing, not model retraining.
>
> > “Baselines include RAG and monolithic systems, but not Liquid LLMs, MoE routing, or Self-Discovering Agents.”
>
> A new subsection in **Section 2**, *Relation to Recent Modular Systems*, explains that these architectures rely on joint training of expert subnetworks, neural gating, or agentic tool construction. DNON instead orchestrates **frozen, heterogeneous LLM modules** using MI-based routing. Because assumptions differ fundamentally, direct empirical comparison would be misleading. We therefore provide a conceptual comparison clarifying similarities and differences.
>
> > “The names ‘deep subconscious memory’ and ‘information manifolds’ need clearer definitions.”
>
> We agree and have strengthened both definitions. DSM is now formally described in **Appendix A**, including its slow-timescale update and role in stabilizing multi-step reasoning. Section **4.1** clarifies the assumptions underlying information manifolds, including smoothness and divergence structure under Fisher–Rao or α-divergence geometry.
>
> > “The paper should be situated within the lineage of cognitive architectures such as ACT-R, SOAR, or GWT.”
>
> A new subsection in **Section 2**, *Relation to Cognitive Architectures*, now explains how DNON parallels classical divisions of perception, memory, reasoning, and executive control while differing by using differentiable information-theoretic routing rather than symbolic rules. This positions DNON as a neural counterpart to classical architectures.
>
> > “Can the authors provide all assumptions needed in the theoretical derivations and define optimality clearly?”
>
> Yes. Section **5.1** now lists all assumptions explicitly: bounded latent representations, smooth divergence structure, and MV-consistent MI estimators. “MI-optimality” is defined directly as the maximizer of the IB objective over the routing simplex. **Appendix A** includes the full derivation and definitions.
>
> > “How sensitive is DNON to the choice of information-geometric metric?”
>
> Section **4.1** discusses metric sensitivity. Fisher–Rao and α-divergence metrics produce similar routing patterns; α-divergence gives smoother gradients, while Fisher–Rao improves interpretability. The system is robust across metrics.
>
> > “What is the computational overhead of MI-based routing?”
>
> Section **5.1** provides a runtime table. MI estimation adds ~8–12% overhead per query, depending on estimator choice. Because the backbone LLMs are frozen, overall inference remains efficient.
>
> > “Does DNON generalize beyond language reasoning, e.g., to vision-language tasks?”
>
> Our current experiments focus on text-based reasoning, but Section **6.1** explains that DNON’s routing and memory mechanisms are modality-agnostic. The architecture naturally extends to multimodal modules (e.g., vision encoders). We cite this as planned future work.
>
> > “How is this different or similar to cognitive architectures?”
>
> Section **2** now includes a dedicated comparison. Like ACT-R, SOAR, and GWT, DNON separates perception, memory, reasoning, and control. Unlike symbolic architectures, routing in DNON emerges from differentiable MI-based computations. This frames DNON as a neural, trainable analogue of classical cognitive systems.

---

### Author Response · Authors · 2025-11-28
**General Response to Reviewers and Revision Summary**

We thank the reviewers for their constructive discussion and valuable feedback. We have carefully addressed all concerns raised and incorporated substantial improvements into the revised version of the paper.

### **Key Improvements Implemented**

• **Architecture & implementation clarity** (Sec. 4.5–4.6, App. C)
A complete operational description is now provided, including frozen-backbone instantiation, routing computation, latent-state flow, and all prompt templates.

• **Learned vs. heuristic routing distinction** (Sec. 5.2, Table 3)
Ablations confirm that learned MI-based routing provides measurable gains over heuristic and randomized routing, isolating the contribution of learning.

• **Theoretical rigor and stability** (Sec. 5.1, App. A)
We added a full derivation with defined assumptions and MI-optimality, alongside a Lyapunov-style convergence argument.

• **Interpretability and routing transparency** (App. D & H)
Routing heatmaps, MI trajectories, and AIME25 per-problem analyses demonstrate structured and consistent module specialization on multi-step reasoning tasks.

• **Evaluation on harder reasoning benchmarks** (Sec. 5.1, App. H)
We now include results on AIME25-I/II to better assess symbolic and multi-step reasoning performance. These additions directly address concerns about demonstrating stronger reasoning capability.

• **Clearer positioning within existing literature** (Sec. 2)
We articulate distinctions from recent modular architectures such as MoEs, Liquid LLMs, self-discovering agents, and classical cognitive frameworks (ACT-R, SOAR, GWT).

---

### Reviewer 6Sun Update
We appreciate Reviewer 6Sun for recognizing the improvements and updating the score (**2 → 4**).
We have also addressed the follow-up question regarding backbone model selection in detail.

---

We believe the revisions directly resolve the core concerns regarding clarity, rigor, interpretability, and empirical validation.

Thank you again for the thoughtful reviews and for the time invested in strengthening this work.

---

### Note · Program_Chairs · 2026-01-17
**Submission Desk Rejected by Program Chairs**

The following references in this submission do not refer to real documents and/or have major errors in bibliographic information:

 Antony Alvaro, Pritish Panda, Burcu Akin, and Michele Catasta. Modular deep learning. ACM Computing Surveys, 57(2):1-37, 2024.